# A UNIFIED CAUSAL VIEW OF INSTRUCTION TUNING

## ABSTRACT

Instruction tuning on a mixture of tasks has improved zero-shot capabilities in natural language processing (NLP) . Nevertheless, existing methods often learn features that exhibit correlations between instruction-formatted samples and target labels, rather than causal relationships. Termed as "spurious correlation" in statistics, such a correlation may change drastically in a new task, making the effect from the learned features to be misleading. To this end, we develop a meta Structural Causal Model (meta-SCM) to integrate different NLP tasks under a single causal structure of the data. Specifically, the meta-SCM introduces multiple latent factors that represent properties of source context language, only some of which causally influence the target labels for a specific task. The key idea is to learn task-required causal factors and only use those to make predictions for a given task. Theoretically, we prove the causal factor can be identified without mixing information from others. Guided by the identifiability, we propose a Structural Instruction Tuning (SIT) method to learn the task-required causal representations that can mimic the causal factors for each task. The utility of our approach is verified by improvements of zero-shot ability on a range of unseen datasets and tasks.

## 1 INTRODUCTION

Pretrained large language models (LLMs) have achieved remarkable success in numerous natural language processing (NLP) tasks (Lewis et al., 2020; Raffel et al., 2020; Brown et al., 2020). Recently, instruction tuning technique has emerged, allowing language models to achieve better generalization on new tasks (Wei et al., 2021a; Sanh et al., 2021; Zhao et al., 2023). In general, instruction tuning reformulates different tasks into the sequence-to-sequence (Seq2Seq) form, with natural language instructions customized for each task. Despite the improved performance of instruction tuning in zero-shot scenarios, current methods fit data by exploiting surface correlations between instruction-formatted samples and target labels, ignoring the invariant data generating process (DGP) underlying the data (Cao et al., 2022). As a result, it may suffer from fragile "spurious correlation" and mislead the prediction, especially when adapting to new tasks (Wang et al., 2022a).

In recent years, structural causal model (SCM) has attracted extensive attention, which describes the underlying DGP, enabling the graphical formalization of causal relationships from observed data (Pearl, 2009a;b; Altman & Krzywinski, 2015; Moraffah et al., 2020; Schölkopf, 2022). Unlike fragile statistical correlations, causal relationships are invariant across domains (Kaur et al., 2022; Zhang et al., 2021a; Sun et al., 2021), which are transferable and highly reliable even when applied to previously unseen domains. To this end, we hope to put the idea of SCM into the practice for instruction tuning to further improve the generalization ability. In this work, we develop a **meta Structural Causal Model (meta-SCM)**, a single causal structure that can integrate different NLP tasks. The causal graph induced by the meta-SCM is depicted on the left in Figure 1.

Specifically, we introduce a set of latent factors to describe the DGP of observed data in NLP tasks. Since datasets used for each task have inherent properties (possibly from sampling bias), the latent factors will be influenced by these dataset properties. As later analyzed, inherent dataset properties cause traditional methods to learn spurious correlations. To address this, we propose identifying and utilizing causal factors for each task. The causal factors for each specific task are a subset of all the latent factors. For example, the connotative semantics in a document are the causal factors for sentiment analysis, while the core denotative semantics are the causal factors for topic classification.

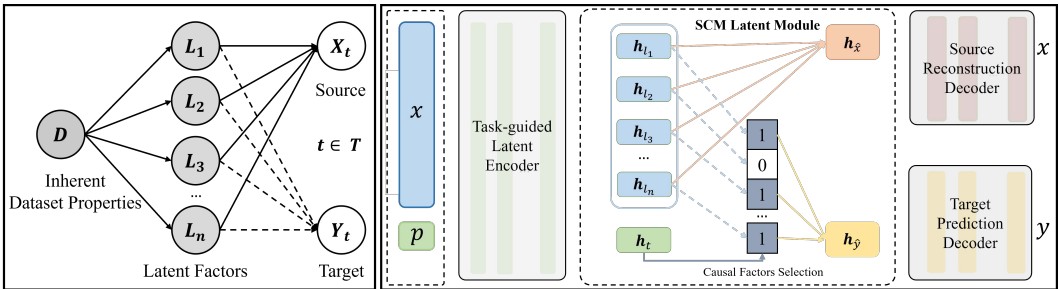

Figure 1: **Left**: The causal graph induced by the meta Structural Causal Model (meta-SCM) for integrating different NLP tasks. White nodes denote observed variables and grey nodes denote unobserved variables. Dashed lines denote edges that may be absent, while solid lines denote invariant processes. See Section 3.1 for a detailed description. **Right**: The model overview of Structural Instruction Tuning (SIT), aiming at learning the representations for task-required causal factors. Task information based on prompts guides the causal factor selection. Detailed description can be found in Section 4.

On the theoretical side, we present the Uniform Identifiability Condition (UIC), a sufficient and necessary condition to ensure identifiability of latent factors in the meta-SCM. The UIC guarantees these factors can be separated without mixing information from other factors by fitting observed data. Importantly, this theoretical result is applicable to a wide range of SCMs with certain topological structures, shedding light on incorporating causality into other areas, such as fairness and debiasing.

On the experimental side, guided by the meta-SCM with identifiability guarantees, we propose a novel **Structural Instruction Tuning (SIT)** method to integrate multiple NLP tasks under the causal view. The key idea is to learn for each task: (1) the causal factor selection mechanisms, (2) the task-required causal representations that can mimic the task-required causal factors, and (3) the causal generative mechanisms to generate the target labels from the causal factors. As shown in the right of Figure 1, the model architecture consists of four components: (i) *SCM Latent Module*, to obtain the causal representation for generating the source and target for each task, where we realize causal factor selection based on a binary adjacency matrix with 0,1 values. (ii) *Task-guided Latent Encoder*, to provide the task-required latent representations of all latent factors for the SCM latent module; (iii) *Source Reconstruction Decoder*, to constrain the latent representations by reconstructing source contexts from all of them; and (iv) *Target Prediction Decoder*, to predict the target label from the causal representation, based on the causal generative mechanism. During testing, the prediction is performed based on the adaptively selected latent factors, according to the learned task-oriented causal generative mechanism.

In summary, the main contributions of this paper are as follows: (i) Theoretically, we provide uniform identifiability conditions based on the topology of SCM. It is innovative that this identifiability results hold across a range of topology structures, rather than being limited to a fixed SCM. (ii) Methodically, we propose a meta-SCM that integrates multiple NLP tasks and introduces a novel tuning method using structural instructions. To the best of our knowledge, it is the first work to capture causal relationships by instruction tuning. (iii) Experimentally, we verify the effectiveness of SIT on both in-domain and out-of-domain (OOD) datasets, e.g., 60.51% improvement on Gigaword in terms of Rouge-L. We also show better cross-task adaptability of SIT on unseen tasks, e.g., 31.30% improvement on RTE in terms of accuracy.

## 2  RELATED WORK

**Causal Representation Learning.** Causal representation learning aims to explore the causal relationships underlying data, by modeling data generating processes rather than merely superficial dependencies (Wang & Jordan, 2021; Schölkopf et al., 2021). This approach has gained significant traction in various areas, including recommendation systems (Zheng et al., 2021; Zhang et al., 2021b; Wei et al., 2021b), computer vision (Niu et al., 2021; Sun et al., 2021; Zhang et al., 2021a), and information retrieval (Joachims et al., 2017), showing significant enhancements in prediction, generalization, and interpretability. In NLP, causal representation learning has been introduced in various tasks such as self-explanation (Liu et al., 2023b), text classification (Qian et al., 2021; Veitch et al., 2021), text summarization (Xie et al., 2021), information extraction (Nan et al., 2021), and

language modeling (Cao et al., 2022). Nevertheless, existing approaches often concentrate on specific NLP tasks, where the typical methodology involves exploring the particular Data Generating Process (DGP) associated with each task and subsequently formulating a corresponding SCM. However, when confronted with multitask scenarios, manually designing a SCM for each task can be laborious. Consequently, the integration of different NLP tasks within a single SCM remains relatively underdeveloped.

**Identifiability Analysis.** The concept of identifiability encompasses whether a representation learned from observed data can match the true underlying latent factors which are responsible for the data generating process, under acceptable transformation operations such as permutation or scaling (Lehmann & Casella, 2006). When incorporating causality into representation learning, it becomes crucial for providing an identifiability guarantee for the latent factors, ensuring that they can be correctly learned. Previous works have analyzed identifiability in various areas, including multimodal (Daunhawer et al., 2023), computer vision (Sun et al., 2021; Lu et al., 2021), and graph neural networks (Chen et al., 2022). However, existing works often follow a paradigm of manually designing a specific Structural Causal Model (SCM), then performing analysis just for that SCM. Consequently, the identifiability results lack generalizability.

**Instruction-based Learning.** Instruction-based learning formulate instances from downstream tasks into instruction-formatted ones, aiming to alleviate the discrepancy between pre-training and fine-tuning (Schick & Schütze, 2021; Liu et al., 2023a). Prompt engineering focuses on constructing effective instructions or prompts, which can be either manually designed (Schick & Schütze, 2021; Puri & Catanzaro, 2019; Petroni et al., 2019; Brown et al., 2020) or automatically searched (Shin et al., 2020; Wang et al., 2022b; Gu et al., 2022; Gao et al., 2021). These instructions are typically composed of discrete words (Shin et al., 2020; Gao et al., 2021), continuous embeddings (Li & Liang, 2021; Lester et al., 2021) or a hybrid of them (Xu et al., 2022; Chen et al., 2023; Liu et al., 2021). Based on proper instruction, the model is able to perform few-shot or zero-shot learning (Gao et al., 2021; Logan IV et al., 2022; Gu et al., 2022; Sun et al., 2022), especially for LLMs (Brown et al., 2020; Wei et al., 2021a; Sanh et al., 2021). However, existing works tend to exploit surface correlations of data (Cao et al., 2022), which may be hard to capture the mapping between instructions and task-required target labels under multitask setting. Thus current multitasking works based on PLM with limited scale often target at special task clusters, e.g., text matching (Xu et al., 2022), text classification (Zhong et al., 2021), knowledge-intensive tasks (Chen et al., 2023).

## 3 PROBLEM FORMULATION AND THEORY

In this section, we formulate various NLP tasks within a single structural causal model, named meta structural causal model (meta-SCM). Furthermore, we propose a uniform identifiability condition (UIC) based on the topological structure of SCM, which guarantees that latent factors in the SCM can be identified without mixing information from others by fitting the observed data.

### 3.1 META STRUCTURAL CAUSAL MODEL

First, we formulate different NLP tasks from a generative perspective. NLP tasks typically require generating target labels, such as summaries, sentiments, or text categories, from source contexts. In our modeling, we abstract the generating process of target labels and source context as a causal graph, as shown in the left of Figure 1. The rationale for this approach is as follow:

- $X_t, Y_t$ represent the source context and target label respectively for NLP tasks, where subscript $t$ denotes a specific task, such as documents and summaries, or sentences and sentiment polarity.

- $L = \{L_1, L_2, L_3, ..., L_n\}$ represents the abstract properties of language which are shared across different tasks. They are unobservable latent factors. The meaning of these properties may refer to linguistic studies (Saussure, 2011), including lexicon, syntax, semantics, and others. Semantics can further be divided into connotative and denotative categories, etc.

- $D$ represents inherent dataset properties.

- $T = \{\mathbf{t_1}, \mathbf{t_2}, \mathbf{t_3}, \cdots, \mathbf{t_m}\}$ represents $m$ different NLP tasks.

- $L \rightarrow X_t, Y_t$ indicates that the source context $X_t$ and target label $Y_t$ are generated from the latent factors $L$. Considering that source context $X_t$ carries all the information of $L$, $X_t$ is pointed by all of $L$. Differently, not all latent factors are used to generate target label $Y_t$ for a certain task.

Consequently, dashed lines are employed to signify the uncertain connection from these latent factors to the target labels $Y_t$.

- $D \to L$ indicates the abstract language properties are influenced by the inherent dataset properties of different datasets across tasks. For instance, news summarization datasets may focus on denotative semantics, while sentiment analysis datasets prioritize connotative elements.

Structural causal models characterize causal relationships between variables through a set of structural functions, which we denote as generating processes in the remainder of the paper. And these structural functions induce a corresponding causal graph. Based on the above causal relations, we formalize the generating process for each variable. Formally, the representations of source contexts can be viewed as random variables $\mathbf{X_t}$, taking values in source observed representation space $\mathcal{X}_t \subseteq \mathbb{R}^{\dim(\mathbf{x_t})}$. Similarly, the representations for target labels and latent factors are random variables $\mathbf{Y_t}$ and $\mathbf{L}$, taking values in target observation representation space $\mathcal{Y}_t \subseteq \mathbb{R}^{\dim(\mathbf{y_t})}$ and latent representation space $\mathcal{L} \subseteq \mathbb{R}^{\dim(\mathbf{l})}$ respectively. Now we formalise the generating process for a given task as follow:

$$\mathbf{L}_i \sim p_{\mathbf{L}_i}(\mathbf{l}_i | \mathbf{d}), \quad \mathbf{X_t} := f_{\mathbf{X_t}}(Pa(\mathbf{X_t})) + \varepsilon_{\mathbf{X_t}}, \quad \mathbf{Y_t} := f_{\mathbf{Y_t}}(Pa(\mathbf{Y_t})) + \varepsilon_{\mathbf{Y_t}}. \tag{1}$$

In Equation 1, $Pa(\cdot)$ denotes the set of parent nodes in the causal graph, indicating that only task-required latent factors will be selected. The symbol $:=$ is often used in SCMs to emphasize that the formula represents a data generating process rather than a simple mathematical equation. Equation 1 is categorized as an additive noise model (ANM), which is a widely used type of SCM. For latent factors $\mathbf{L}_i$, considering that exponential family distribution has universal approximation capability for a given distribution, we assume the prior of latent factors $p_{\mathbf{L}_i}(\mathbf{L}_i = \mathbf{l}_i | \mathbf{D} = \mathbf{d})$ is given by :

$$p_{\mathbf{L}_i}(\mathbf{l}_i | \mathbf{d}) = \prod_{j=1}^{\dim(\mathbf{l}_i)} \frac{Q_{i,j}(l_{i,j})}{Z_{i,j}(\mathbf{d})} \exp\left[ \sum_{k=1}^{\dim(\mathbf{T}_{i,j})} T_{i,jk}(l_{i,j}) \lambda_{i,jk}(\mathbf{d}) \right]. \tag{2}$$

For exponential family distribution (Equation 2), we adopt the common used notation. Upperclass letters denote functions not random variables. $Z_i$ are called partition functions, serving to normalization. $Q_i$ are the base measures, $\mathbf{T}_i$ are sufficient statistics and $\lambda_i$ are the corresponding parameters.

**Spurious Correlation.** Due to inherent properties in the training dataset, probably from sampling bias, the target labels exhibit spurious correlation with non-causal latent factors. For example, in a dataset sampled from pizza enthusiasts for sentiment analysis, pizza, as a food concept, will co-occur with positive emotion frequently, causing spurious correlation between food and sentiment labels. **Causally speaking, there exist backdoor paths between the target labels and non-causal latent factors through the inherent dataset properties $D$.** Existing methods that simply use the whole source text for prediction, and thus include both causal and non-causal information, suffer from this spurious correlation. To address this issue, an effective approach involves identifying the causal latent factors for the task and utilizing only those to predict target labels. In the following, we will first provide a theoretical guarantee for the latent factors' identifiability (Sec. 3.2). Based on this, we propose finding the latent factors required for different tasks by maximizing likelihood (Sec. 4).

## 3.2 Uniform Identifiability Condition

As discussed previously, identifiability plays an essential role in our methodology. **In this section, we present theoretical results guaranteeing identifiability of the latent factors.** Specifically, we propose a novel Uniform Identifiability Condition (UIC), a sufficient and necessary condition to determine whether an SCM is identifiable. The term "uniform" signifies that this condition can effectively apply to a wide range of structural causal models (SCMs) with specific topological structures, rather than being restricted to a particular SCM.

**Identifiability.** Intuitively, identifiability means that the latent factors can be learned without any information mixing. Following, we present the formal definition of identifiability:

**Definition 1** (Identifiability). *True latent factors* $[\mathbf{L}_1, \mathbf{L}_2, \cdots, \mathbf{L}_n]$ *are* $\sim_P$ *identifiable to learning latent factors* $[\tilde{\mathbf{L}}_1, \tilde{\mathbf{L}}_2, \cdots, \tilde{\mathbf{L}}_n] = [\tilde{f}_{\mathbf{X_t}}^{-1}(\mathbf{X_t})_{\mathbf{L}_1}, \tilde{f}_{\mathbf{X_t}}^{-1}(\mathbf{X_t})_{\mathbf{L}_2}, \cdots, \tilde{f}_{\mathbf{X_t}}^{-1}(\mathbf{X_t})_{\mathbf{L}_n}]$ *if the following condition is met:*

$$\forall i \in \{1, 2, \cdots, n\}, \quad \tilde{\mathbf{T}}_i(\tilde{\mathbf{L}}_i) = P_i \mathbf{T}_i(\mathbf{L}_i) + \mathbf{b}_i. \tag{3}$$

In Equation 3, $\mathbf{T}_i$ denotes sufficient statistics of $\mathbf{L}_i$, $P_i$ denotes a permutation matrix, $\mathbf{b}_i$ is a vector. Intuitively, Equation 3 means that the difference between true latent factor $\mathbf{L}_i$ and learning latent

factors $\tilde{\mathbf{L}}_i$ is no more than a permutation transformation with a linear shift on there sufficient statistics. Besides, this transformation preserves the individuality of each latent factor and ensures that there is no information mixing between them.

Previous works have theoretically shown that it is impossible to identify the true latent factors without any assumptions for the data generating process (Hyvärinen & Pajunen, 1999). In this work, we adopt follow mild assumptions which are commonly used in other identifiability works (Khemakhem et al., 2020; Sun et al., 2021; Lu et al., 2021):

**Assumption 1** (**Bijective**). *The generating functions $f_{\mathbf{X}_t}$, $f_{\mathbf{Y}_t}$ are bijective.*

**Assumption 2** (**Denoising**). *Characterisitic functions of $\varepsilon_{\mathbf{X}_t}$, $\varepsilon_{\mathbf{Y}_t}$ are nonzero almost everywhere.*

**Assumption 3** (**Transformation**). *The sufficient statistics $\mathbf{T}$ are linear independent on every nonzero measure subset of $L$ and are differentiable almost everywhere.*

**Assumption 4** (**Variety**). *The number of different datasets, with differing inherent properties $D$, be $n_D \geq n_0 = \max(dim(\mathbf{l}_i) \times dim(\mathbf{T}_{i,j})) + 1$, and the following matrix has full column rank:*

$$\mathbf{H}_{\mathbf{t}} = [\boldsymbol{\lambda}(\mathbf{d}_1) - \boldsymbol{\lambda}(\mathbf{d}_0), \boldsymbol{\lambda}(\mathbf{d}_2) - \boldsymbol{\lambda}(\mathbf{d}_0), ..., \boldsymbol{\lambda}(\mathbf{d}_{n_0}) - \boldsymbol{\lambda}(\mathbf{d}_0)]. \quad (4)$$

The meaning of each assumptions are detailed explained in Appendix A.1.

$\sim_P$ **identifiable for SCM.** An SCM is $\sim_P$ identifiable if all the true latent factors in the SCM are $\sim_P$ identifiable to the learning latent factors. **We propose a uniform identifiability condition, which guarantees that our meta-SCM, as described in Section 3.1, is $\sim_P$ identifiable.**

**Theorem 1.** *Considering the data generating process described in Section 3.1, where $\mathbf{X}_t$, $\mathbf{Y}_{t,t \in \{t_1, t_2, \cdots, t_m\}}$ are generated according to Equation 1, and $\mathbf{L}_{i,i \in \{1,2,\cdots,n\}}$ has the distribution specified in Equation 2, as well as the fulfillment of Assumptions 1 - 4. We introduce a set of sets $\mathcal{F}$ that describes the topology structure of a SCM and can be used to determine whether the SCM is identifiable. $\mathcal{F}$ is generated by the following steps:*

   *1. $\emptyset$, $Pa(\mathbf{X}_{t_1})$, $Pa(\mathbf{Y}_{t_1})$, $\cdots$, $Pa(\mathbf{X}_{t_m})$, $Pa(\mathbf{Y}_{t_m}) \in \mathcal{F}$*

   *2. Set $\mathcal{A}$, $\mathcal{B} \in \mathcal{F} \Rightarrow$ Set $\mathcal{A} - \mathcal{B}$, $\mathcal{B} - \mathcal{A} \in \mathcal{F}$. Here $\mathcal{A} - \mathcal{B} = \mathcal{A} \cap \bar{\mathcal{B}}$*

*The SCM is $\sim_P$ identifiable if the set of sets $\mathcal{F}$ includes all singleton sets $\mathbf{L}_i$, that is*

$$\{\mathbf{L}_1\}, \{\mathbf{L}_2\}, \cdots, \{\mathbf{L}_n\} \in \mathcal{F}.$$

**Proof sketch.** (1) Fourier transformation is applied to the marginal probability density equation to eliminate noise in DGP. Considering latent factors follow exponential family distributions, we take logarithms on both sides of the equations, converting it to additive form. (2) Notice that the set $\mathcal{F}$ expands gradually through the application of the "set subtraction" operator. Consequently, performing the subtraction operator on the additive form equations yields new equations with fewer latent factors. The condition that $\mathcal{F}$ contains all singleton sets implies that all the latent factors can finally be separated in their corresponding equation. (Detailed proofs are provided in Appendix A.2)

**We then prove the condition is not only sufficient but also necessary,** as stated in Theorem 2.

**Theorem 2.** *Considering the data generating process described in Section 3.1, we employ a binary adjacency matrix denoted as $A$ to represent the topology relations between $\mathbf{L}_i$ and $\mathbf{Y}_t$. The matrix $A$ comprises $m$ rows and $n$ columns, where $m$ is the number of $\mathbf{Y}_t$, and $n$ is the number of latent factors $\mathbf{L}_i$. Specifically, a value of 1 at position $(i, j)$ indicates that $\mathbf{L}_j$ has a direct effect on $\mathbf{Y}_i$, while a value of 0 indicates no direct effect. Latent factors in a SCM are identifiable if and only if the following equation holds. We refer to the equation as the **uniform identifiability condition (UIC).***

$$\mathbb{1}\left(\frac{1}{m}\left[A^\top A + (1-A)^\top(1-A)\right] - I_{n \times n}\right) = 0_{n \times n}. \quad (5)$$

*In Equation 5, $\mathbb{1}(\cdot)$ is an indicator function, which defined as $[\mathbb{1}(A)]_{ij} = \begin{cases} 0, & 0 \leq a_{ij} < 1 \\ 1, & a_{ij} = 1 \end{cases}$*

**Proof sketch.** (1) We propose a criterion: For any two distinct latent factors $\mathbf{L}_i$ and $\mathbf{L}_j$ in the SCM, their child sets (i.e., sets containing $\mathbf{X}_t$ and $\mathbf{Y}_t$ pointed by $\mathbf{L}$) are not identical. We then prove that

a negative answer to this criterion implies non-identifiability of the SCM, **which represents the contrapositive form of the necessary condition for identifiability.** (2) We establish the equivalence between the criterion and the sufficient result proposed in Theorem 1. (3) Combining the results (1) and (2), we conclude that both the condition in Theorem 1 and the criterion serve as necessary and sufficient conditions for determining the identifiability of a SCM. (4) Expressing these results in matrix form, yielding Equation 5. Detailed proofs and lemmas are provided in Appendix A.3, A.4.

# 4 METHOD

Guided by above theory, we propose a Structural Instruction Tuning (SIT) method which induces causality into instruction tuning, so that the model can explicitly capture the causal relationships among tasks, source contexts and target labels.[1] The key idea is to learn (1) the representations for task-required causal factors, and (2) the task-oriented causal generative mechanisms for generating target labels given causal factors. In the following, we first introduce the proposed model architecture (Sec. 4.1), followed by the learning and inference process (Sec. 4.2).

## 4.1 MODEL ARCHITECTURE

As shown in the right of Figure 1, the model architecture consists of four components: (i) *SCM Latent Module*, to realize the DGP in the meta-SCM; (ii) *Task-guided Latent Encoder*, to learn the representations of latent factors extracted from source sequences with the guidance of tasks; (iii) *Source Reconstruction Decoder*, to constrain the latent representations by reconstructing source sequences from all of them; and (iv) *Target Prediction Decoder*, to predict the target sequence from the automatically selected causal representations of specific tasks.

**SCM Latent Module**. This module consists of causal factor selection $(\mathbf{h}_t, \mathbf{h}_l \rightarrow \mathbf{h}_l^t)$ and combination $(\mathbf{h}_l \rightarrow \mathbf{h}_{\hat{x}}$ and $\mathbf{h}_l^t \rightarrow \mathbf{h}_{\hat{y}})$. Here $\mathbf{h}_t$ denotes the representations of task variable $T$, $\mathbf{h}_l = \{\mathbf{h}_{l_1}, \mathbf{h}_{l_2}, ..., \mathbf{h}_{l_n}\}$ denotes the representations of all the latent factors $L$, and $\mathbf{h}_l^t$ denotes the representations of task-required causal latent factors selected from $L$.

- **Causal Factor Selection**. We assume all the latent factors are causal for $x$, while not all are causal for $y$. Since the causal factors are difficult to define manually, we design a task-guided selection mechanism to pick them automatically. Specifically, we encode hybrid prompts to obtain task representation $\mathbf{h}_t$, and map it into a latent mask vector $\mathbf{m}_t$ of length $n$ (see Appendix C). Then $\mathbf{m}_t$ selects the task-required causal factors through element-wise multiplication, i.e., $\mathbf{h}_l^t = \mathbf{m}_t \otimes \mathbf{h}_l$.

- **Causal Factor Combination**. We merge all latent representations to obtain the guidance for generating $x$ by linear combination of $\mathbf{h}_{l_1}, \mathbf{h}_{l_2}, ..., \mathbf{h}_{l_n}$, i.e., $\mathbf{h}_{\hat{x}} = \mathbf{W}_1 \mathbf{h}_l + \mathbf{b}_1$. Similarly, we linear combine task-required causal representations $\mathbf{h}_l^t$ to obtain the guidance for $y$, i.e., $\mathbf{h}_{\hat{y}} = \mathbf{W}_2 \mathbf{h}_l^t + \mathbf{b}_2$.

**Task-guided Latent Encoder**. To obtain representations $\mathbf{h}_l$ in the SCM latent module, we learn the inference model $q_\chi(l|x, t)$ implemented by a task-guided latent encoder $\chi$, which maps source sequences $x$ into the latent factors $l$ with the guidance of task variable $t$. Firstly, we utilize a Transformer encoder $\mathrm{Trm}_{\mathrm{enc}}$ as the base encoder to encode the prompted input into the latent space, i.e., $\{\mathbf{h}_s, \mathbf{h}_p, \mathbf{h}_x, \mathbf{h}_e\} = \mathrm{Trm}_{\mathrm{enc}}(\{[s], p, x, [e]\})$, where $p$ are prompts, and $[s]/[e]$ are special tokens representing the start and end of the input sequence respectively. Following Karpukhin et al. (2020); Tang et al. (2021), we extract $\mathbf{h}_s$ as task-guided source representations $\mathbf{h}_{x^t} \in \mathbb{R}^{d_h}$. Then, we perform linear transformation to map $\mathbf{h}_{x^t}$ into a set of latent representations, i.e., $\mathbf{h}_l = \mathbf{W}_3 \mathbf{h}_{x^t} + \mathbf{b}_3$.

**Source Reconstruction Decoder**. To reconstruct the source sequence $x$ with the guidance $\mathbf{h}_{\hat{x}}$ from the SCM latent module, we learn the causal generative mechanisms $p_\theta(x|l)$ for source generation by a source reconstruction decoder $\theta$. Firstly, we utilize a Transformer decoder $\mathrm{Trm}_{\mathrm{dec}}$ to obtain the output latent space of $x$. Following Xia et al. (2020), for the decoder input, we replace its first embedding with $\mathbf{h}_{\hat{x}} \in \mathbb{R}^{d_h}$, i.e., $\mathbf{o}_0 = \mathrm{Trm}_{\mathrm{dec}}(\mathbf{h}_{\hat{x}})$, for it is essential to guide the generation of the subsequent tokens. For the decoder output, we add $\mathbf{h}_{\hat{x}}$ to the hidden states to obtain the final latent space $\{\hat{\mathbf{o}}_i\}_{i=1}^{|x|}$ of $x$, i.e., $\hat{\mathbf{o}}_i = \mathbf{h}_{\hat{x}} + \mathbf{o}_i$. After that, a language modeling layer calculates the vocabulary selection probability for generating $x$, i.e., $P_{x_i} = \mathrm{LM}(\hat{\mathbf{o}}_i)$.

**Target Prediction Decoder**. To predict the target sequence $y$ with the guidance $\mathbf{h}_{\hat{y}}$ from the SCM latent module, we learn the causal generative mechanisms $p_\sigma(y|l)$ for target generation by a target

---

[1]Note that both the source and target are formulated into text sequences, with hybrid prompts as instructions to indicate task information (see Appendix B), which is also necessary for new tasks.

prediction decoder $\sigma$. Similar to source sequence reconstruction, we incorporate $\mathbf{h}_{\hat{y}}$, the combined representation of selected causal factors, into this prediction decoder composed of a Transformer decoder and a LM layer.

## 4.2 Learning and Inference

By learning causal representations and task-oriented causal generative mechanisms, the model can adaptively select the task-required causal factors to perform a specific task during testing.

**Learning Method**. During the training stage, we propose four loss function as follow.

- **Source Reconstruction Loss** is applied to constrain the latent representations by maximizing the likelihood of the source sequence $x$, i.e., $\mathcal{L}_{rec} = -\mathbb{E}[\log p_\theta(x|l)]$.

- **Target Prediction Loss** is applied to guide the target sequence prediction by maximizing the likelihood of the target sequence $y$, i.e., $\mathcal{L}_{pre} = -\mathbb{E}[\log p_\sigma(y|l)]$.

- **UIC Loss** is employed to enforce the identifiability of latent factors $l$, which **incorporates the theoretical results into our model**. It stems from the UIC condition (Theorem 2), with derivation details provided in the Appendix D. The loss function can be expressed as:

$$\mathcal{L}_{uic} = \frac{1}{m^\alpha} \sum_{i,j=1}^{n} \left[ \sum_{k=1}^{m} a_{ki}a_{kj} + (1-a_{ki})(1-a_{kj}) - m\delta_{ij} \right]^\alpha, \tag{6}$$

  where $\alpha$ is a hyperparameter, $\delta$ is the Kronecker delta function defined as: $\delta_{ij} = \begin{cases} 0, & i \neq j \\ 1, & i = j \end{cases}$

- **Task Distinction Loss** is employed to guarantee diverse causal factor selection for different tasks. It penalizes the scenarios where different tasks rely on the same set of causal factors. Drawing inspiration from the UIC loss, the loss function is designed as:

$$\mathcal{L}_{dis} = \frac{1}{n^\alpha} \sum_{k,k'=1}^{m} \left[ \sum_{i=1}^{n} a_{ik}a_{ik'} + (1-a_{ik})(1-a_{ik'}) - n\delta_{kk'} \right]^\alpha. \tag{7}$$

Overall, we combine four learning objectives described above as the final loss function, i.e.,

$$\mathcal{L} = \mathcal{L}_{rec} + \mathcal{L}_{pre} + \lambda_{uic}\mathcal{L}_{uic} + \lambda_{dis}\mathcal{L}_{dis}, \tag{8}$$

where $\mathcal{L}_{uic}$ and $\mathcal{L}_{dis}$ are regularization terms we call *causal factor constraint*.

**Inference and Prediction**. During testing, we predict the target label for given samples $x$ from task $t$ based on the learned inference model $q_\chi(l|x,t)$ and the causal generative mechanism $p_\sigma(y|l)$. Task instructions enable our model to adaptively select the task-required causal factors, even for new tasks.

## 5 Experimental Settings

**Tasks and Datasets**. We collect 21 datasets from seven classical NLP tasks including generation and classification tasks, as shown in Table 1. To evaluate cross-task adaptability, LA and NLI datasets are fully held out, unseen during training. For the other five tasks, some datasets are sampled to construct the mixed training set following Wei et al. (2021a); Raffel et al. (2020), while some are reserved as held-out datasets to test out-of-domain (OOD) performance on training mixture tasks. The details of task selection and the sample strategy refer to Appendix E.

**Baselines**. To verify the superiority of SIT, we adopt several baselines for comparison: (i) Multi-task Fine-Tuning (**MFT**) trains our backbone PLM on the multitask training dataset without instructions. (ii) Vanilla Instruction Tuning (**Vanilla-IT**) apply textual-instruction tuning for the PLM.

**Model architecture**. SIT is applicable for all the Seq2Seq models[2]. Considering computational efficiency, we use $\text{BART}_{large}$[3] for model initialization, which has hidden size $d_h = 1024$ and the vocabulary size $d_v = 50265$. The additional complexity is from the source reconstruction decoder (254M) and the SCM latent module (30M), and only the latter one is involved during inference.

---

[2]The code will be made available upon publication.

[3]https://huggingface.co/facebook/bart-large

Table 1: Datasets for different tasks. Some are included in the training set, while some are held-out.

| | Task | Training-mixture datasets | Held-out datasets |
|---|---|---|---|
| **Training mixture** | Summarization (SUM) | XSUM, CNNDM | Gigaword |
| | Reading Comprehension (RC) | Duorc$_{self}$, Duorc$_{para}$ | Squad |
| | Topic Classification (TC) | AG, Trec | DBPedia |
| | Paraphrase Detection (PD) | PAWS | MRPC, QQP |
| | Sentiment Analysis (SA) | IMDB, Yelp | SST-2 |
| **Held-out** | Linguistic Acceptability (LA) | | CoLA |
| | Natural Language Inference (NLI) | | QNLI, RTE, WNLI, MNLI$_m$,MNLI$_{mm}$ |

Table 2: In-domain performance over the test set of training-mixture. Best results are marked bold.

| Method | SUM | | RC | | TC | | PD | SA | |
|---|---|---|---|---|---|---|---|---|---|
| | XSUM | CNNDM | Duorc$_{self}$ | Duorc$_{para}$ | AG | Trec | PAWS | IMDB | Yelp |
| MFT | 31.41 | 26.01 | 34.92 | 23.92 | 87.78 | 80.8 | 42.88 | 90.57 | 67.28 |
| Vanilla-IT | 30.92 | 36.67 | 54.81 | 31.94 | 90.67 | 65.0 | 55.79 | 95.17 | 70.32 |
| SIT | **33.59** | **38.33** | **63.27** | **38.30** | **91.75** | **85.4** | **76.75** | **96.11** | **78.44** |

**Training details**. We utilize Adam optimizer with learning rate of 3e-5 and weight decay of 0.01. Also, we apply warm-up over the first 10% steps. The batch size is 256 and the total training steps is about 10k. We train on one NVIDIA Tesla V100-32GB GPUs for about 3 days. We set the max length of source sequences as 550, and that of target sequences as 64. We set $m = 5$ and select the best $n$ from 4 to 16. $\lambda_{uic}$ and $\lambda_{dis}$ are selected from 0 to 1. More details refer to Appendix F.

**Evaluation details**. Similar to the previous work (Wei et al., 2021a; Sanh et al., 2021), We use **ROUGE-L** as the evaluation metric for SUM and RC tasks and **accuracy** for other tasks. We evaluate over the test set for Duorc and all the datasets of SUM and TC tasks, or the dev set for other tasks.

## 6 EXPERIMENTAL RESULTS

We target five research questions (RQ) as follows: **(RQ1)** How does SIT perform on in-domain datasets? **(RQ2)** How does SIT perform on out-of-domain datasets? **(RQ3)** How is the cross-task adaptability of SIT to unseen tasks? **(RQ4)** How is the few-shot learning capability of SIT? **(RQ5)** What is the impact of causal factor constraint? Accordingly, we organize the following experiments.

**In-domain Performance**. To answer **RQ1**, we first compare models on the training mixture task listed in Table 1 under the consistent training setting. We evaluate the performance on their test set, as shown in Table 2. We make 3 key observations: (i) On 7 out of 9 datasets, Vanilla-IT outperforms MFT, indicating the importance of textual instructions to provide task information. (ii) On all the datasets, SIT outperforms Vanilla-IT in terms of all the metrics, suggesting that SIT with structural instructions is better at capturing task specifications while learning the shared knowledge. (iii) On the PAWS from PD task, SIT outperforms Vanilla-IT significantly, which is 37.57% better in terms of accuracy. The possible reason for the larger margin than other datasets is that the number of training samples for PD is limited (see Appendix E) and it is harder for Vanilla-IT to learn the mapping between the instruction-formatted samples and target labels, while SIT can handle it.

**Out-of-domain Performance**. To answer **RQ2**, we compare models over the held-out datasets of training tasks. The results are shown in the left part of Table 3, from which we have 3 observations. (i) Vanilla-IT outperforms MFT on 5 out of 6 datasets, indicating the adaptability of instruction tuning. (ii) SIT outperforms Vanilla-IT on 5 out of 6 datasets. For example, SIT outperforms Vanilla-IT 60.51% on Gigaword in terms of Rouge-L. This result demonstrates that SIT capturing the stable causal relationships of each task helps model deal with the domain transfer, while Vanilla-IT only capturing superficial correlation performs worse. (iii) Models perform badly on the DBPedia dataset, for the possible reason that its target categories are totally different from the seen datasets. By conducting case study, we find that some of the sample topics could be grouped into semantically similar ones, e.g., classifying "Company" as "Business", where "Business" are seen during training.

Table 3: OOD and cross-task performance of held-out datasets. Best results are marked bold.

| Method | OOD Performance | | | | | | Cross-task Performance | | | | |
| | SUM | RC | TC | PD | | SA | LA | NLI | | | | |
| | Gigaword | Squad | DBPedia | MRPC | QQP | SST-2 | CoLA | $MNLI_m$ | $MNLI_{mm}$ | QNLI | RTE | WNLI |
|---|---|---|---|---|---|---|---|---|---|---|---|---|
| MFT | 0.43 | 59.34 | 0.20 | 63.97 | 17.14 | 80.20 | 0.00 | 15.83 | 18.35 | 6.94 | 11.82 | 19.72 |
| Vanilla-IT | 12.51 | 61.87 | 0.50 | 31.62 | 63.19 | 92.43 | 0.00 | 31.82 | 31.85 | 50.52 | 47.29 | **56.34** |
| SIT | **20.08** | **70.29** | 0.50 | **70.1** | **68.24** | **93.23** | 0.00 | **39.62** | **39.89** | **55.96** | 62.09 | **56.34** |

Table 4: Ablation study for the UIC loss and task distinction loss on zero-shot performance.

| Method | SUM | RC | PD | | SA | NLI | | | | |
| | Gigaword | Squad | MRPC | QQP | SST-2 | $MNLI_m$ | $MNLI_{mm}$ | QNLI | RTE | WNLI |
|---|---|---|---|---|---|---|---|---|---|---|
| SIT | **20.08** | **70.29** | **70.1** | **68.24** | **93.23** | **39.62** | **39.89** | **55.96** | **62.09** | **56.34** |
| w/o $\mathcal{L}_{uic}$ | 19.57 | 69.59 | 31.62 | 63.18 | 92.89 | 35.28 | 35.10 | 44.28 | 52.35 | 54.93 |
| w/o $\mathcal{L}_{dis}$ | 19.14 | 69.04 | 31.62 | 63.19 | 91.74 | 31.82 | 31.85 | 50.45 | 47.29 | 43.66 |

**Cross-task Generalization**. To answer **RQ3**, we compare the cross-task adaptability of models on held-out tasks listed in Table 1 under the zero-shot settings. The results are shown in the right part of Table 3, from which we have 3 observations. (i) Vanilla-IT outperforms MFT significantly, indicating the cross-task generalization ability of instruction tuning. (ii) SIT further achieves better results than baselines on 4 out of 6 datasets of new tasks. For example, SIT outperforms Vanilla-IT 31.30% on RTE in terms of accuracy. It indicates that by introducing latent factors as structural bridge between the textual instructions and the target labels, SIT can boost the understanding of human instructions, which enables model to have better generalizability when transferring to new tasks. (iii) Models perform the worst on the CoLA dataset. Similar to DBPedia, CoLA also has a totally different target categories (i.e., acceptable, unacceptable) from seen tasks.

**Few-shot Learning Capability**. To answer **RQ4**, we further evaluate the few-shot performance for both OOD datasets and unseen tasks. Concretely, we randomly pick 1024 samples from training set for each task to fine-tune trained models. As shown in Figure 2, SIT can achieve a noticeable boost (69.03%) which is twice of the gain (38.16%) of Vanilla-IT, even for the new task CoLA. This result indicates the better learning capability of SIT based on structural instructions rather than mere textural instructions. More few-shot results refer to Appendix G.

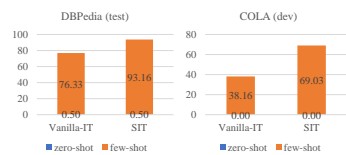

Figure 2: Few-shot results.

**Impact of Causal Factor Constraint**. To answer **RQ5**, we conduct ablation studies by setting $\lambda_{uic}$ and $\lambda_{dis}$ as 0 to remove the UIC loss and task distinction loss, respectively. From Table 4 we can observe that: (i) Training without the UIC loss harms zero-shot performance, indicating that holding UIC indeed helps to migrate spurious correlations and boost the generalization ability. (ii) Training without the task distinction loss will also hurt the performance, indicating that realizing the different selection of causal factors can help to learn the task-oriented causal generative mechanism.

## 7 CONCLUSIONS

This paper has explored a novel instruction tuning method based on a meta-SCM proposed to integrate different NLP tasks. By incorporating the SCM, SIT enables model to capture the underlying causal relationships and has better adaptability to deal with domain transfer and new tasks. Besides, we provide a uniform identifiability conditions and apply it to regularize the learning process. We hope our paper will inspire further research on incorporating the causality into the algorithm design.

A limitation of SIT is the dependence on prescribed human instructions to distinguish different tasks. We will attempt self-learning from raw data in the future. Besides the preliminary exploration for conventional NLP tasks, we are interested in more complex tasks like reasoning, which may require sophisticated SCM designs. The practice combined with LLMs also deserves further exploration, where strong parametric knowledge may need to be taken into account when designing SCM.

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

# APPENDIX for "A Unified Causal View of Instruction Tuning"

OVERVIEW:

- Appendix A contains the detailed data generating process, detailed proofs for all theoretical results in the main paper, as well as the proposed lemmas.

- Appendix B contains the prompt engineering to indicate task information.

- Appendix C contains the details of causal factor selection, including the encoding of the task representations and the mapping into latent mask vectors.

- Appendix D contains the details of causal factor constraint, including the key idea of the UIC Loss and the implementation of matrix $A$ from the Theorem 2.

- Appendix E contains the details of tasks and datasets, including task selection and sampling strategy.

- Appendix F contains additional details of training and inference process.

- Appendix G contains additional experimental results under few-shot learning.

## A   LEMMAS AND PROOFS

This section is structured as follows. We first provide some notations employed in this paper. In Appendix A.1, we provide a more detailed description for the data generating process in the main paper. Appendix A.2 presents the complete proof of Theorem 1. In Appendix A.3, we propose and prove useful lemmas that will be utilized in the proof of Theorem 2. Finally, Appendix A.4 offers the full proof of Theorem 2.

**Notations.** In this section, we adhere to a uniform notation scheme as in the main paper. Random variables are denoted by uppercase letters, while specific values are represented by lowercase letters, unless specified otherwise. For instance, $\mathbf{X}$ is a random variable and $\mathbf{x}$ is a particular value. Vector values are indicated by bold typeface (e.g., $\mathbf{x}$), while scalar values are represented using regular typeface (e.g., $x$). Additionally, calligraphic-style letters are used to denote representation spaces. For example, $\mathcal{X}$ represents a representation space where $\mathbf{x}$ belongs, with $\mathbf{x} \in \mathcal{X} \subseteq \mathbb{R}^{\dim(\mathbf{x})}$.

### A.1   DATA GENERATING PROCESS

Before presenting the lemmas and proofs for identifiability, it is crucial to provide a comprehensive explanation of the data generating process. Understanding the data generating process is pivotal in the study of causality, as it unveils the causal mechanisms (denoted as assignment functions in Section 3.1) through which observed variables are produced by latent factors. In this regard, we employ the structural causal model (SCM), a widely utilized framework, to describe the data generating process. Formally, let $\mathbf{x_t} \in \mathbb{R}^{\dim(\mathbf{x_t})}, \mathbf{y_t} \in \mathbb{R}^{\dim(\mathbf{y_t})}, \mathbf{l}_i \in \mathbb{R}^{\dim(\mathbf{l}_i)}$. The parent set of $\mathbf{X_t}$ denoted as $Pa(\mathbf{X_t})$ and the parent set of $\mathbf{Y_t}$ denoted as $Pa(\mathbf{Y_t})$. As explained in Section 3.1, the source context $\mathbf{X_t}$ carries all the information of $\mathbf{L}$, hence $Pa(\mathbf{X_t}) = \{\mathbf{L}_1, \mathbf{L}_2, \mathbf{L}_3, \cdots, \mathbf{L}_n\}$. In order to simplify the expression of exponential family distribution, we define $\Theta_{\mathbf{x_t}} \triangleq \{f_{\mathbf{x_t}}, \mathbf{\Phi}_{\mathbf{x_t}}\}$, where $f_{\mathbf{x_t}}$ denotes the invertible generating function, $\mathbf{\Phi}_{\mathbf{x_t}}$ represents the set of sufficient statistics $\mathbf{T}$ and it's coefficient $\boldsymbol{\lambda}$.

The joint probability density of source context $\mathbf{X_t}$ and latent factors $\mathbf{L}_i$ can be written as:

$$p_{\Theta_{\mathbf{x_t}}}(\mathbf{x_t}, Pa(\mathbf{x_t})|\mathbf{d}) = p_{\Theta_{\mathbf{x_t}}}(\mathbf{x_t}, Pa(\mathbf{x_t})|\mathbf{d}) \tag{9}$$

$$= p_{f_{\mathbf{x_t}}}(\mathbf{x_t}|Pa(\mathbf{x_t})) \cdot p_{\mathbf{\Phi}_{\mathbf{x_t}}}(Pa(\mathbf{x_t}|\mathbf{d}). \tag{10}$$

According to the additive noise model (ANM) assumption (Equation 1), the data generating process of $\mathbf{x_t}$ can be written as:

$$\mathbf{x_t} = f_{\mathbf{x_t}}(Pa(\mathbf{x_t})) + \varepsilon_{\mathbf{x_t}}, \quad \varepsilon_{\mathbf{x_t}} \sim p_\varepsilon(\varepsilon). \tag{11}$$

Using Equation 11, we can rewrite Equation 9 as:

$$p_{\Theta_{\mathbf{x_t}}}(\mathbf{x_t}, Pa(\mathbf{x_t})|\mathbf{d}) = p_{f_{\mathbf{x_t}}}(\mathbf{x_t}|Pa(\mathbf{x_t})) \cdot p_{\Phi_{\mathbf{x_t}}}(Pa(\mathbf{x_t})|\mathbf{d}) \tag{12}$$

$$\Rightarrow p_{\Theta_{\mathbf{x_t}}}(\mathbf{x_t}, Pa(\mathbf{x_t})|\mathbf{d}) = p_{\varepsilon_{\mathbf{x_t}}}(\mathbf{x_t} - f_{\mathbf{x_t}}(Pa(\mathbf{x_t}))) \cdot p_{\Phi_{\mathbf{x_t}}}(Pa(\mathbf{x_t})|\mathbf{d}). \tag{13}$$

Considering that exponential family has universal approximation capability for probability density function, we assume the conditional probability density function $p_{\Phi_{\mathbf{x_t}}}(Pa(\mathbf{x_t})|\mathbf{d})$ is given by:

$$p_{\Phi_{\mathbf{x_t}}}(Pa(\mathbf{x_t})|\mathbf{d}) = \prod_{i=1}^{n} p_{\mathbf{T}_i, \boldsymbol{\lambda}_i}(\mathbf{l}_i|\mathbf{d}) \tag{14}$$

$$\Rightarrow p_{\Phi_{\mathbf{x_t}}}(Pa(\mathbf{x_t})|\mathbf{d}) = \prod_{i=1}^{n} \prod_{j=1}^{\dim(\mathbf{l_i})} p_{\mathbf{T}_i, \boldsymbol{\lambda}_i}(l_{i,j}|\mathbf{d}) \tag{15}$$

$$\Rightarrow p_{\Phi_{\mathbf{x_t}}}(Pa(\mathbf{x_t})|\mathbf{d}) = \prod_{i=1}^{n} \prod_{j=1}^{\dim(\mathbf{l_i})} \frac{Q_{i,j}(l_{i,j})}{Z_{i,j}(\mathbf{d})} \exp\left[\sum_{k=1}^{\dim(\mathbf{T}_{i,j})} T_{i,jk}(l_{i,j})\lambda_{i,jk}(\mathbf{d})\right]. \tag{16}$$

Notice that we employ a slightly different notation, $p_{\mathbf{T}_i, \boldsymbol{\lambda}_i}(\mathbf{l}_i|\mathbf{d})$, instead of $p_{\mathbf{L}_i}(\mathbf{l}_i|\mathbf{d})$, to denote the conditional probability density of the latent factor $\mathbf{l}_i$, which is aimed at emphasizing that the latent factors are represented using exponential family distributions.

Equation 16 is called exponential family distribution, where $Q_{i,j}$ is the base measure, $Z_{i,j}$ is the partition function, i.e. normalization function, $T_{i,jk}$ is one of the sufficient statistics and $\lambda_{i,jk}$ is the corresponding coefficient. We can also rewrite $T_{i,jk}$ and $\lambda_{i,jk}$ in vector form:

$$\mathbf{T}_{i,j}(l_{i,j}) = [T_{i,j1}(l_{i,j}), T_{i,j2}(l_{i,j}), \cdots, T_{i,jk}(l_{i,j})]^T. \tag{17}$$

$$\boldsymbol{\lambda}_{i,j}(\mathbf{d}) = [\lambda_{i,j1}(\mathbf{d}), \lambda_{i,j2}(\mathbf{d}), \cdots, \lambda_{i,jk}(\mathbf{d})]^T. \tag{18}$$

Substituting it in Equation 16:

$$p_{\Phi_{\mathbf{x_t}}}(Pa(\mathbf{x_t})|\mathbf{d}) = \prod_{i=1}^{n} \prod_{j=1}^{\dim(\mathbf{l_i})} \frac{Q_{i,j}(l_{i,j})}{Z_{i,j}(\mathbf{d})} \exp\left[\boldsymbol{\lambda}_{i,j}(\mathbf{d})^\top \mathbf{T}_{i,j}(l_{i,j})\right]. \tag{19}$$

In this work, we adopt the following mild assumptions for the data generating processes, which are commonly used in other works(Khemakhem et al., 2020; Sun et al., 2021; Lu et al., 2021):

**Assumption 1** (**Bijective**). *The generating functions $f_{\mathbf{X_t}}$, $f_{\mathbf{Y_t}}$ are bijective.*

**Assumption 2** (**Denoising**). *Characterisitic functions of $\varepsilon_{\mathbf{X_t}}$, $\varepsilon_{\mathbf{Y_t}}$ are nonzero almost everywhere.*

**Assumption 3** (**Transformation**). *The sufficient statistics $\mathbf{T}$ are linear independent on every nonzero measure subset of $L$ and are differentiable almost everywhere.*

**Assumption 4** (**Variety**). *The number of different datasets, with differing inherent properties $D$, be $n_D \geq n_0 = \max(dim(\mathbf{l}_i) \times dim(\mathbf{T}_{i,j})) + 1$, and the following matrix has full column rank:*

$$\mathbf{H_t} = [\boldsymbol{\lambda}(\mathbf{d}_1) - \boldsymbol{\lambda}(\mathbf{d}_0), \boldsymbol{\lambda}(\mathbf{d}_2) - \boldsymbol{\lambda}(\mathbf{d}_0), ..., \boldsymbol{\lambda}(\mathbf{d}_{n_0}) - \boldsymbol{\lambda}(\mathbf{d}_0)]. \tag{4}$$

Note that Assumption 1 is commonly used in identifiability works. Assumption 2 is generally satisfied for most continuous random variables, including Gaussian, exponential, and beta distributions. By applying Fourier transformation, this assumption helps eliminate the effect of noise in Equation 1. Assumption 3 is satisfied for all distributions belonging to the strongly exponential distribution family. Assumption 4 stipulates that the training datasets should contain a sufficient number of different datasets, and the full column rank of $\mathbf{H_t}$ indicates that datasets should be diverse enough.

### A.2 PROOF OF THEOREM 1

**Theorem 1.** *Considering the data generating process described in Section 3.1, where $\mathbf{X_t}$, $\mathbf{Y}_{\mathbf{t}, \mathbf{t} \in \{\mathbf{t_1}, \mathbf{t_2}, \cdots, \mathbf{t_m}\}}$ are generated according to Equation 1, and $\mathbf{L}_{i, i \in \{1, 2, \cdots, n\}}$ has the distribution specified in Equation 2, as well as the fulfillment of Assumptions 1 - 4. We introduce a set of sets $\mathcal{F}$ that describes the topology structure of a SCM and can be used to determine whether the SCM is identifiable. $\mathcal{F}$ is generated by the following steps:*

1. $\emptyset$, $Pa(\mathbf{X_{t_1}})$, $Pa(\mathbf{Y_{t_1}})$, $\cdots$, $Pa(\mathbf{X_{t_m}})$, $Pa(\mathbf{Y_{t_m}}) \in \mathcal{F}$

2. Set $\mathcal{A}$, $\mathcal{B} \in \mathcal{F} \Rightarrow$ Set $\mathcal{A} - \mathcal{B}$, $\mathcal{B} - \mathcal{A} \in \mathcal{F}$. Here $\mathcal{A} - \mathcal{B} = \mathcal{A} \cap \bar{\mathcal{B}}$

*The SCM is $\sim_P$ identifiable if the set of sets $\mathcal{F}$ includes all singleton sets $\mathbf{L}_i$, that is*

$$\{\mathbf{L}_1\}, \{\mathbf{L}_2\}, \cdots, \{\mathbf{L}_n\} \in \mathcal{F}.$$

*Proof.* The proof of the theorem can be roughly divided into two main steps. First, we transform the equations of probability density into an additive form. This step allows us to express the equations as a sum of individual components. Second, we apply the subtraction operator to the additive form equations, yielding equations with fewer latent factors. Consequently, each final equation contains only one of the latent factors.

**Step 1. Transforming** We begin our proof by stating that the learning marginal probability density on $\mathbf{X_t}$ and $\mathbf{Y_t}$ equals the true marginal probability density. For source context $\mathbf{X}_t$:

$$p_{\Theta_{\mathbf{x_t}}}(\mathbf{x_t}) = p_{\tilde{\Theta}_{\mathbf{x_t}}}(\mathbf{x_t}) \tag{20}$$

$$\Rightarrow p_{f_{\mathbf{x_t}}, \Phi_{\mathbf{x_t}}}(\mathbf{x_t}|\mathbf{d}) = p_{\tilde{f}_{\mathbf{x_t}}, \tilde{\Phi}_{\mathbf{x_t}}}(\mathbf{x_t}|\mathbf{d}) \tag{21}$$

$$\Rightarrow \int p_{f_{\mathbf{x_t}}}(\mathbf{x_t}|Pa(\mathbf{x_t})) p_{\Phi_{\mathbf{x_t}}}(Pa(\mathbf{x_t})|\mathbf{d}) \prod_{i=1}^{n} dl_{\mathbf{i}}$$

$$= \int p_{\tilde{f}_{\mathbf{x_t}}}(\mathbf{x_t}|Pa(\mathbf{x_t})) p_{\tilde{\Phi}_{\mathbf{x_t}}}(Pa(\mathbf{x_t})|\mathbf{d}) \prod_{i=1}^{n} dl_{\mathbf{i}} \tag{22}$$

$$\Rightarrow \int p_{\varepsilon_{\mathbf{x_t}}}(\mathbf{x_t} - f_{\mathbf{x_t}}(Pa(\mathbf{x_t}))) p_{\Phi_{\mathbf{x_t}}}(Pa(\mathbf{x_t})|\mathbf{d}) \prod_{i=1}^{n} dl_{\mathbf{i}}$$

$$= \int p_{\varepsilon_{\mathbf{x_t}}}(\mathbf{x_t} - \tilde{f}_{\mathbf{x_t}}(Pa(\mathbf{x_t}))) p_{\tilde{\Phi}_{\mathbf{x_t}}}(Pa(\mathbf{x_t})|\mathbf{d}) \prod_{i=1}^{n} dl_{\mathbf{i}} \tag{23}$$

$$\Rightarrow \int p_{\varepsilon_{\mathbf{x_t}}}(\mathbf{x_t} - \bar{\mathbf{x}}_{\mathbf{t}}) p_{\Phi_{\mathbf{x_t}}}(f_{\mathbf{x_t}}^{-1}(\bar{\mathbf{x}}_{\mathbf{t}})|\mathbf{d}) \left| \det(J_{f_{\mathbf{x_t}}^{-1}}(\bar{\mathbf{x}}_{\mathbf{t}})) \right| d\bar{\mathbf{x}}_{\mathbf{t}}$$

$$= \int p_{\varepsilon_{\mathbf{x_t}}}(\mathbf{x_t} - \bar{\mathbf{x}}_{\mathbf{t}}) p_{\tilde{\Phi}_{\mathbf{x_t}}}(\tilde{f}_{\mathbf{x_t}}^{-1}(\bar{\mathbf{x}}_{\mathbf{t}})|\mathbf{d}) \left| \det(J_{\tilde{f}_{\mathbf{x_t}}^{-1}}(\bar{\mathbf{x}}_{\mathbf{t}})) \right| d\bar{\mathbf{x}}_{\mathbf{t}} \tag{24}$$

$$\Rightarrow \int p_{\varepsilon}(\mathbf{x_t} - \bar{\mathbf{x}}_{\mathbf{t}}) p_{\Phi_{\mathbf{x_t}}, f_{\mathbf{x_t}}, \mathbf{t}}(\bar{\mathbf{x}}_{\mathbf{t}}) d\bar{\mathbf{x}}_{\mathbf{t}} = \int p_{\varepsilon}(\mathbf{x_t} - \bar{\mathbf{x}}_{\mathbf{t}}) p_{\tilde{\Phi}_{\mathbf{x_t}}, \tilde{f}_{\mathbf{x_t}}, \mathbf{t}}(\bar{\mathbf{x}}_{\mathbf{t}}) d\bar{\mathbf{x}}_{\mathbf{t}} \tag{25}$$

$$\Rightarrow (p_{\varepsilon_{\mathbf{x_t}}} * p_{\Phi_{\mathbf{x_t}}, f_{\mathbf{x_t}}, \mathbf{t}})(\mathbf{x_t}) = (p_{\varepsilon_{\mathbf{x_t}}} * p_{\tilde{\Phi}_{\mathbf{x_t}}, \tilde{f}_{\mathbf{x_t}}, \mathbf{t}})(\mathbf{x_t}) \tag{26}$$

$$\Rightarrow F[p_{\varepsilon_{\mathbf{x_t}}}](\omega) F[p_{\Phi_{\mathbf{x_t}}, f_{\mathbf{x_t}}, \mathbf{t}}](\omega) = F[p_{\varepsilon_{\mathbf{x_t}}}](\omega) F[p_{\tilde{\Phi}_{\mathbf{x_t}}, \tilde{f}_{\mathbf{x_t}}, \mathbf{t}}](\omega) \tag{27}$$

$$\Rightarrow F[p_{\Phi_{\mathbf{x_t}}, f_{\mathbf{x_t}}, \mathbf{t}}](\omega) = F[p_{\tilde{\Phi}_{\mathbf{x_t}}, \tilde{f}_{\mathbf{x_t}}, \mathbf{t}}](\omega) \tag{28}$$

$$\Rightarrow p_{\Phi_{\mathbf{x_t}}, f_{\mathbf{x_t}}, \mathbf{t}}(\mathbf{x_t}) = p_{\tilde{\Phi}_{\mathbf{x_t}}, \tilde{f}_{\mathbf{x_t}}, \mathbf{t}}(\mathbf{x_t}). \tag{29}$$

From Equation 21 to Equation 22, we introduce variables $Pa(\mathbf{x_t})$ into the formula and integrate them. This step is a commonly used technique to incorporate target variables in probability density equations. In Equation 24, the symbol $J$ represents the Jacobian matrix, while $|\det|$ denotes the generalized determinant of the matrix, $\det|A| = \sqrt{\det(A^\top A)}$. In Equation 25, we introduce $p_{\Phi_{\mathbf{x_t}}, f_{\mathbf{x_t}}, \mathbf{t}}(\bar{\mathbf{x}}_{\mathbf{t}}) = p_{\tilde{\Phi}_{\mathbf{x_t}}}(\tilde{f}_{\mathbf{x_t}}^{-1}(\bar{\mathbf{x}}_{\mathbf{t}})|\mathbf{d}) \left| \det(J_{\tilde{f}_{\mathbf{x_t}}^{-1}}(\bar{\mathbf{x}}_{\mathbf{t}})) \right|$ for convenience. It is obviously that the Equation 25 is in the form of convolution. In Equation 26, $F$ means Fourier transformation which is a useful tool to simplify convolution. From Equation 26 to Equation 28, we make an assumption that the characteristic function of noise $F[p_{\varepsilon}]$ is non-zero almost everywhere, hence this term can be eliminated. Finally, we acquire the denoised result. Then taking the logarithm on the both sides of Equation 29 and

substituting the $p_{\mathbf{\Phi}_{\mathbf{x_t}}}$ with the exponential family distribution, we have

$$
\begin{aligned}
&\log\left|\det(J_{f_{\mathbf{x_t}}^{-1}}(\mathbf{x_t}))\right| \\
&+ \sum_{i=1}^{n}\sum_{j=1}^{\dim(\mathbf{l}_i)}\left(Q_{i,j}\left(\left[f_{\mathbf{x_t}}^{-1}(\mathbf{x_t})\right]_{i,j}\right) - Z_{i,j}(\mathbf{d}) + \sum_{k=1}^{\dim(\mathbf{T}_{i,j})} T_{i,jk}\left(\left[f_{\mathbf{x_t}}^{-1}(\mathbf{x_t})\right]_{i,j}\right)\lambda_{i,jk}(\mathbf{d})\right) \\
&= \log\left|\det(J_{\tilde{f}_{\mathbf{x_t}}^{-1}}(\mathbf{x_t}))\right| \\
&+ \sum_{i=1}^{n}\sum_{j=1}^{\dim(\mathbf{l}_i)}\left(\tilde{Q}_{i,j}\left(\left[\tilde{f}_{\mathbf{x_t}}^{-1}(\mathbf{x_t})\right]_{i,j}\right) - \tilde{Z}_{i,j}(\mathbf{d}) + \sum_{j=1}^{\dim(\mathbf{T}_{i,j})} \tilde{T}_{i,jk}\left(\left[\tilde{f}_{\mathbf{x_t}}^{-1}(\mathbf{x_t})\right]_{i,j}\right)\tilde{\lambda}_{i,jk}(\mathbf{d})\right).
\end{aligned}
\tag{30}
$$

Notice that we have sufficient different tasks or datasets $\mathbf{t}$, that is, there exits $\dim(\mathbf{l}_i)\times\dim(\mathbf{T}_{i,j})+1$ different $t$. Pluging these different $\mathbf{t}$ in Equation 30 resulting to $\dim(\mathbf{l}_i)\times\dim(\mathbf{T}_{i,j})+1$ equations. By subtracting the first equation from the second equation up to the last equation, we obtain a set of equations indexed by $l = 1, 2, \dots, \dim(\mathbf{l}_i)\times\dim(\mathbf{T}_{i,j})$:

$$
\begin{aligned}
&\sum_{i}^{n}\left[\langle\mathbf{T}_i\left(\left[f_{\mathbf{x_t}}^{-1}(\mathbf{x_t})\right]_i\right), \overline{\boldsymbol{\lambda}_i}(\mathbf{d}_l)\rangle + \sum_{j}\log\frac{Z_{i,j}(\mathbf{d}_0)}{Z_{i,j}(\mathbf{d}_l)}\right] \\
&= \sum_{i}^{n}\left[\langle\tilde{\mathbf{T}}_i\left(\left[\tilde{f}_{\mathbf{x_t}}^{-1}(\mathbf{x_t})\right]_i\right), \overline{\tilde{\boldsymbol{\lambda}}_i}(\mathbf{d}_l)\rangle + \sum_{j}\log\frac{\tilde{Z}_{i,j}(\mathbf{d}_0)}{\tilde{Z}_{i,j}(\mathbf{d}_l)}\right].
\end{aligned}
\tag{31}
$$

In Equation 31, we define $\overline{\boldsymbol{\lambda}_i}(\mathbf{d}_l) = \boldsymbol{\lambda}_i(\mathbf{d}_l) - \boldsymbol{\lambda}_i(\mathbf{d}_0)$. In order to simplified Equation 31 further, we define $\mathbf{w}_{l,i} = \sum_j \frac{\tilde{Z}_{i,j}(\mathbf{d}_0)Z_{i,j}(\mathbf{d}_l)}{\tilde{Z}_{i,i}(\mathbf{d}_l)Z_{i,j}(\mathbf{d}_0)}$. Then we rewrite these equations in matrix form:

$$
\sum_{i}^{n}\mathbf{H}_{\mathbf{d}}^{i,\top}\mathbf{T}_i\left(\left[f_{\mathbf{x_t}}^{-1}(\mathbf{x_t})\right]_i\right) = \sum_{i}^{n}\tilde{\mathbf{H}}_{\mathbf{t}}^{i,\top}\tilde{\mathbf{T}}_i\left(\left[\tilde{f}_{\mathbf{x_t}}^{-1}(\mathbf{x_t})\right]\right) + \mathbf{w}_{l,i},
\tag{32}
$$

where $\mathbf{H}_{\mathbf{d}}^i = [\boldsymbol{\lambda}_i(\mathbf{d}_1) - \boldsymbol{\lambda}_i(\mathbf{d}_0), \boldsymbol{\lambda}_i(\mathbf{d}_2) - \boldsymbol{\lambda}_i(\mathbf{d}_0), ..., \boldsymbol{\lambda}_i(\mathbf{d}_{n_0}) - \boldsymbol{\lambda}_i(\mathbf{d}_0)]$, $n_0 = \dim(\mathbf{l}_i) \times \dim(\mathbf{T}_{i,j})$.

**Step 2. Separation** Similar to $\mathbf{x_t}$, we can express the transformed equations for $\mathbf{y_t}$ as well. Notice that the parent sets of $\mathbf{x_t}$ encompass all latent factors $\mathbf{l}_i$, while the parent sets of $\mathbf{y_t}$ usually encompass a subset of latent factors $\mathbf{l}_i$. We use the notation $idx(Pa(\mathbf{Y_t}))$ to represent the indices of the latent factors comprising the set $Pa(\mathbf{Y_t})$. We obtain m transformed equations for each $\mathbf{Y_{t_s}}$, $s = 1, 2, 3, \cdots, m$:

$$
\sum_{i\in idx(Pa(\mathbf{Y_{t_s}}))}\mathbf{H}_{\mathbf{t}}^{i,\top}\mathbf{T}_i\left(\left[f_{\mathbf{y_{t_i}}}^{-1}(\mathbf{y_{t_s}})\right]_i\right) = \sum_{i\in idx(Pa(\mathbf{Y_{t_s}}))}\tilde{\mathbf{H}}_{\mathbf{t}}^{i,\top}\tilde{\mathbf{T}}_i\left(\left[\tilde{f}_{\mathbf{y_{t_s}}}^{-1}(\mathbf{y_{t_s}})\right]\right) + \mathbf{w}_{l,i}.
\tag{33}
$$

Furthermore, it is crucial to note that the latent factors $\mathbf{l}_i$ are shared by $\mathbf{Y_t}$. Based on this property, we can express the transformed equations for the pair of target variables $(\mathbf{y_{t_s}}, \mathbf{y_{t_{s'}}})$ as follows:

$$
\sum_{\substack{i\in idx(Pa(\mathbf{Y_{t_s}})\cup \\ Pa(\mathbf{Y_{t_{s'}}}))}}\mathbf{H}_{\mathbf{t}}^{i,\top}\mathbf{T}_i\left(\left[f_{\mathbf{y_{t_i}}}^{-1}(\mathbf{y_{t_s}}, \mathbf{y_{t_{s'}}})\right]_i\right) = \sum_{\substack{i\in idx(Pa(\mathbf{Y_{t_s}})\cup \\ Pa(\mathbf{Y_{t_{s'}}}))}}\tilde{\mathbf{H}}_{\mathbf{t}}^{i,\top}\tilde{\mathbf{T}}_i\left(\left[\tilde{f}_{\mathbf{y_{t_s}}}^{-1}(\mathbf{y_{t_s}}, \mathbf{y_{t_{s'}}})\right]\right) + \mathbf{w}_{l,i}.
\tag{34}
$$

Notice that the subtraction of the two sets satisfies the following equation:

$$
\mathcal{A} - \mathcal{B} \triangleq \mathcal{A} \cap \bar{\mathcal{B}} = (\mathcal{A}\cap\bar{\mathcal{B}})\cup(\mathcal{B}\cap\bar{\mathcal{B}}) = (\mathcal{A}\cup\mathcal{B})\cap\bar{\mathcal{B}} = (\mathcal{A}\cup\mathcal{B}) - \mathcal{B}.
\tag{35}
$$

Due to the inclusion property $\mathcal{B}\subset\mathcal{A}\cup\mathcal{B}$, the expression $(\mathcal{A}\cup\mathcal{B}) - \mathcal{B}$ represents the removal of identical elements from the set $\mathcal{A}\cup\mathcal{B}$ that are also present in $\mathcal{B}$. It is noteworthy that this type of set

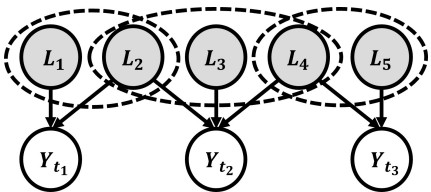

Figure 3: Identifiable latent factors.

subtraction demonstrates a striking similarity to algebraic subtraction. In parallel with the expansion of the set of sets $\mathcal{F}$ through set subtraction, we can utilize algebraic subtraction on Equations 33 and Equations 34 to derive new equations that involve fewer latent factors. Given the condition that D encompasses all singleton sets, it follows that all the latent factors can ultimately be isolated in their respective equations, as shown below:

$$\mathbf{H}_{\mathbf{t}}^{i,\top}\mathbf{T}_i\left(\left[f_{\mathbf{y}_{\mathbf{t}_i}}^{-1}(\mathbf{y}_{\mathbf{t_s}})\right]_i\right) = \tilde{\mathbf{H}}_{\mathbf{t}}^{i,\top}\tilde{\mathbf{T}}_i\left(\left[\tilde{f}_{\mathbf{y}_{\mathbf{t_s}}}^{-1}(\mathbf{y}_{\mathbf{t_s}})\right]\right) + \mathbf{w}_{l,i},$$
$$i \in \{1, 2, \cdots, n\}, \quad \mathbf{t_s} \in \{\mathbf{t_1}, \mathbf{t_2}, \cdots, \mathbf{t_m}\}. \tag{36}$$

Notice that the matrix $\mathbf{H}_{\mathbf{t}}^i$ has full rank, we multiply it's inverse matrix on both sides of Equation 36:

$$\mathbf{T}_i\left(\left[f_{\mathbf{y}_{\mathbf{t}_i}}^{-1}(\mathbf{y}_{\mathbf{t_s}})\right]_i\right) = \mathbf{M}_{\mathbf{t}}^{i,\top}\tilde{\mathbf{T}}_i\left(\left[\tilde{f}_{\mathbf{y}_{\mathbf{t_s}}}^{-1}(\mathbf{y}_{\mathbf{t_s}})\right]\right) + \mathbf{v}_{l,i},$$
$$i \in \{1, 2, \cdots, n\}, \quad \mathbf{t_s} \in \{\mathbf{t_1}, \mathbf{t_2}, \cdots, \mathbf{t_m}\}, \tag{37}$$

where $\mathbf{M}_{\mathbf{t}}^i = (\mathbf{H}_{\mathbf{t}}^{i,\top})^{-1}\tilde{\mathbf{H}}_{\mathbf{t}}^{i,\top}$, $\mathbf{v}_{l,i} = (\mathbf{H}_{\mathbf{t}}^{i,\top})^{-1}\mathbf{w}_{l,i}$.

Finally, we will prove that the matrix $\mathbf{M}_{\mathbf{t}}^i$ is a permutation matrix, demonstrating the $\sim_P$ identifiability of the SCM. We adopt the method from Khemakhem et al. (2020) for this proof. Firstly, we consider the matrix $\mathbf{T}$. Under Assumption 4, the Jacobian of $\mathbf{T}_i$ has a full column rank $n$, implying that the Jacobian of $\mathbf{T}_i(f^{-1})$ is also of rank $n$. Consequently, the matrix $\mathbf{M}_{\mathbf{t}}^i$ is also of rank $n$. Secondly, we analyze two cases based on the dimension $k$ of the sufficient statistics: (1) $k = 1$; (2) $k > 1$. In the case of $k = 1$, the matrix $\mathbf{T}_i$ becomes an $n \times n$ square matrix. Since $\mathbf{T}i$ has a full rank, the matrix $\mathbf{M}\mathbf{t}^i$ is also of full rank, indicating its invertibility. In the case of $k > 1$, we can directly apply Lemma 3 from Khemakhem et al. (2020) to prove the invertibility of $\mathbf{M}_{\mathbf{t}}^i$. Lastly, assuming that both $f$ and the sufficient statistics $\mathbf{T}_i$ are twice differentiable, we apply Theorem 2 and Theorem 3 from Khemakhem et al. (2020) to demonstrate that $\mathbf{M}_{\mathbf{t}}^i$ is a permutation matrix. $\qquad \square$

**Intuition.** To provide an intuitive understanding of Theorem 1, we present an identification process for Figure 3. Initially, we consider $\mathbf{Y}_{\mathbf{t}_1}$, which is pointed by $\mathbf{L}_1$ and $\mathbf{L}_2$. Solely relying on the information from $\mathbf{Y}_{\mathbf{t}_1}$ can not identify these latent factors. Next, we incorporate $\mathbf{Y}_{\mathbf{t}_2}$ into the analysis. By leveraging the information of $\mathbf{Y}_{\mathbf{t}_2}$, we can identify $\mathbf{L}_1$ and $\mathbf{L}_2$, for $\mathbf{L}_1$ exclusively points to $\mathbf{Y}_{\mathbf{t}_1}$, while $\mathbf{L}_2$ points to both $\mathbf{Y}_{\mathbf{t}_1}$ and $\mathbf{Y}_{\mathbf{t}_2}$. Subsequently, we include $\mathbf{Y}_{\mathbf{t}_3}$ in our analysis. Following the same procedure as before, the remaining three latent factors can be identified.

### A.3 LEMMAS

Before presenting the complete proof of Theorem 2, we first provide several useful lemmas.

**Lemma 1.** *Considering the data generating process described in Section 3.1. If there exist two distinct latent factors $\mathbf{L}_i$ and $\mathbf{L}_j$ such that their child sets $Ch(\mathbf{L}_i)$ and $Ch(\mathbf{L}_j)$ are identical, i.e., $Ch(\mathbf{L}_i) = Ch(\mathbf{L}_j)$, then $\mathbf{L}_i$ and $\mathbf{L}_j$ can not be identified.*

*Proof.* We begin the proof with the equation of joint probability density:

$$p(\mathbf{X}_{\mathbf{t_1}}, \mathbf{Y}_{\mathbf{t_1}}, \cdots, \mathbf{X}_{\mathbf{t_m}}, \mathbf{Y}_{\mathbf{t_m}}|\mathbf{d})$$
$$= p(\mathbf{X}_{\mathbf{t_1}}, \mathbf{Y}_{\mathbf{t_1}}, \cdots, \mathbf{X}_{\mathbf{t_m}}, \mathbf{Y}_{\mathbf{t_m}}|\mathbf{L}_1, \mathbf{L}_2, \cdots, \mathbf{L}_n) \cdot p(\mathbf{L}_1, \mathbf{L}_2, \cdots, \mathbf{L}_n|\mathbf{d}) \tag{38}$$

$$= \prod_{\mathbf{t}=\mathbf{t_1}}^{\mathbf{t_m}} p(\mathbf{X}_{\mathbf{t}}|\mathbf{L}_1, \mathbf{L}_2, \cdots, \mathbf{L}_n) \cdot \prod_{\mathbf{t}=\mathbf{t_1}}^{\mathbf{t_m}} p(\mathbf{Y}_{\mathbf{t}}|Pa(\mathbf{Y}_{\mathbf{t}})) \cdot p(\mathbf{L}_1, \mathbf{L}_2, \cdots, \mathbf{L}_n|\mathbf{d}). \tag{39}$$

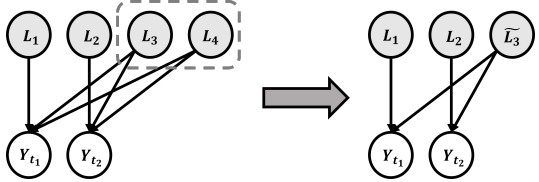

Figure 4: Unidentifiable latent factors.

We denoted $Ch(\mathbf{L}_i) = Ch(\mathbf{L}_j) \triangleq Ch$, $Ch = \{\mathbf{X_{t_1}}, \mathbf{X_{t_2}}, \cdots, \mathbf{X_{t_m}}, \mathbf{Y_{t'_1}}, \cdots, \mathbf{Y_{t'_q}}\}$, in which $\{\mathbf{Y_{t'_1}}, \cdots, \mathbf{Y_{t'_q}}\} \subseteq \{\mathbf{Y_{t_1}}, \mathbf{Y_{t_2}}, \cdots, \mathbf{Y_{t_m}}\}$.

Back to the Equation 39,

$$p(\mathbf{X_{t_1}}, \mathbf{Y_{t_1}}, \cdots, \mathbf{X_{t_m}}, \mathbf{Y_{t_m}}|\mathbf{d})$$

$$= \prod_{\mathbf{t}=\mathbf{t_1}}^{\mathbf{t_m}} p(\mathbf{X_t}|\mathbf{L}_1, \mathbf{L}_2, \cdots, \mathbf{L}_n) \cdot \prod_{\mathbf{t}=\mathbf{t_1}}^{\mathbf{t_m}} p(\mathbf{Y_t}|Pa(\mathbf{Y_t})) \cdot p(\mathbf{L}_1, \mathbf{L}_2, \cdots, \mathbf{L}_n|\mathbf{d}) \tag{40}$$

$$= \prod_{\mathbf{t}=\mathbf{t_1}}^{\mathbf{t_m}} p(\mathbf{X_t}|(\mathbf{L}_1, \mathbf{L}_2, \cdots, \mathbf{L}_{-i}, \mathbf{L}_{-j}, \cdots, \mathbf{L}_n), (\mathbf{L}_i, \mathbf{L}_j))$$

$$\cdot \prod_{\mathbf{t}\in\{\mathbf{t'_1}, \cdots, \mathbf{t'_q}\}} p(\mathbf{Y_t}|(Pa(\mathbf{Y_t}), \mathbf{L}_{-i}, \mathbf{L}_{-j}), (\mathbf{L}_i, \mathbf{L}_j)) \cdot \prod_{\mathbf{t}\notin\{\mathbf{t'_1}, \cdots, \mathbf{t'_q}\}} p(\mathbf{Y_t}|Pa(\mathbf{Y_t})) \tag{41}$$

$$\cdot p((\mathbf{L}_1, \mathbf{L}_2, \cdots, \mathbf{L}_{-i}, \mathbf{L}_{-j}, \cdots, \mathbf{L}_n), (\mathbf{L}_i, \mathbf{L}_j)|\mathbf{d}). \tag{42}$$

Note that $\mathbf{L}_i$ and $\mathbf{L}_j$ always appear together in a term. Considering the following transformation:

$$(\mathbf{L}_i, \mathbf{L}_j) \rightarrow (\mathbf{L}'_i, \mathbf{L}'_j), \quad n = \min\left(\left\lceil \frac{\dim(\mathbf{L}_i)}{2} \right\rceil, \left\lceil \frac{\dim(\mathbf{L}_j)}{2} \right\rceil\right) \tag{43}$$

$$\mathbf{L}'_i = \begin{cases} \mathbf{L}'_{i[1:n]} = \mathbf{L}_{j[1:n]} \\ \mathbf{L}'_{i[n+1:\dim(\mathbf{L}_i)]} = \mathbf{L}_{i[n+1:\dim(\mathbf{L}_i)]} \end{cases}, \quad \mathbf{L}'_j = \begin{cases} \mathbf{L}'_{j[1:n]} = \mathbf{L}_{i[1:n]} \\ \mathbf{L}'_{j[n+1:\dim(\mathbf{L}_i)]} = \mathbf{L}_{j[n+1:\dim(\mathbf{L}_i)]} \end{cases} \tag{44}$$

The purpose of this transformation is to interchange the 1st to nth dimensions of $\mathbf{L}_i$ and $\mathbf{L}_j$. As a result, the transformed variables $\mathbf{L}'_i$ and $\mathbf{L}'_j$ incorporate the information from both $\mathbf{L}_i$ and $\mathbf{L}_j$. Note that both the original pair $(\mathbf{L}_i, \mathbf{L}_j)$ and the transformed pair $(\mathbf{L}'_i, \mathbf{L}'_j)$ satisfy Equation 39, indicating that it is impossible to uniquely recover the original pair $(\mathbf{L}_i, \mathbf{L}_j)$ without information mixing. Consequently, $\mathbf{L}_i$ and $\mathbf{L}_j$ are not identifiable. □

**Intuition.** Figure 4 provides an intuitive understanding of Lemma 1. As depicted in Figure 4, when two latent factors $\mathbf{L_3}$ and $\mathbf{L_4}$ share the same child set $\{\mathbf{Y_{t_1}}, \mathbf{Y_{t_2}}\}$, it is equivalent to considering these two latent factors as a single variable.

**Lemma 2.** *Assuming the number of observed variables $\mathbf{Y}$ is m, if the number of hidden variables $\mathbf{Z}$ is greater than $2^m - 1$, then the causal graph is unidentifiable.*

*Proof.* Lemma 2 can be derived straightforwardly from Lemma 1. The number of different non-empty subsets of $\{\mathbf{Y_{t_1}}, \mathbf{Y_{t_2}}, \cdots, \mathbf{Y_{t_m}}\}$ is given by

$$\sum_i^m C_m^i = C_m^1 + C_m^2 + \cdots + C_m^m = 2^m - 1. \tag{45}$$

□

**Intuition.** Although the proof for Lemma 2 is technically straightforward, its meaning is quite interesting. Intuitively, Lemma 2 highlights the necessity of an adequate number of observed variables to identify latent factors. In this work, these observed variables correspond to distinct tasks or diverse datasets.

**Lemma 3.** *Assuming the number of observed variables* $\mathbf{Y}$ *is* $m$, *for any observed variables* $\mathbf{Y}_{\mathbf{t_i}}$, *its parent set satisfies the following:*

$$|Pa(\mathbf{Y}_{\mathbf{t_i}})| \leq 2^{m-1}. \tag{46}$$

*In Equation 46, the notation* $|A|$ *denotes the cardinality of a set* $A$. *For a finite set, the cardinality represents the number of elements it contains.*

*Proof.* We present a proof by contradiction. Let us assume that the given condition is violated, i.e., $|Pa(\mathbf{Y}_{\mathbf{t_i}})| \geq 2^{m-1} + 1$, which implies that there are at least $2^{m-1} + 1$ latent factors $\mathbf{L}$ pointing to $\mathbf{Y}_{\mathbf{t_i}}$. Considering that all the child sets of these latent factors contain $\mathbf{Y}_{\mathbf{t_i}}$, the only difference lies in the remaining $m - 1$ latent factors. According to Lemma 2, the number of different child sets is limited to $2^{m-1} - 1 + 1 = 2^{m-1}$ (including the empty set). However, the parent set $Pa(\mathbf{Yt_i})$ contains at least $2^{m-1} + 1$ latent factors, indicating that there must exist two different latent factors with the same child set. This contradicts the initial assumption of identifiable latent factors. Consequently, we conclude that the condition $|Pa(\mathbf{Y}_{\mathbf{t_i}})| \leq 2^{m-1}$ holds. $\square$

**Lemma 4.** *Considering a set of sets* $\mathcal{F}$ *that describes the topology structure of a SCM.* $\mathcal{F}$ *is generated by the following steps:*

1. *$\emptyset$, $Pa(\mathbf{X}_{\mathbf{t_1}})$, $Pa(\mathbf{Y}_{\mathbf{t_1}})$, $\cdots$, $Pa(\mathbf{X}_{\mathbf{t_m}})$, $Pa(\mathbf{Y}_{\mathbf{t_m}}) \in \mathcal{F}$*

2. *Set $\mathcal{A}$, $\mathcal{B} \in \mathcal{F} \Rightarrow$ Set $\mathcal{A} - \mathcal{B}$, $\mathcal{B} - \mathcal{A} \in \mathcal{F}$. Here $\mathcal{A} - \mathcal{B} = \mathcal{A} \cap \bar{\mathcal{B}}$*

*The set of sets* $\mathcal{F}$ *includes all singleton sets* $\mathbf{L}_i$, *that is* $\{\mathbf{L}_1\}, \{\mathbf{L}_2\}, \cdots, \{\mathbf{L}_n\} \in \mathcal{F}$, **if and only if** *($\Leftrightarrow$), For any two distinct latent factors* $\mathbf{L}_i$ *and* $\mathbf{L}_j$ *in the SCM, their child sets are not identical.*

*Proof.* We will first prove the direction "$\Rightarrow$" (i.e., "only if"). We present a proof by contradiction. Let us assume that there exists two distinct latent factors $\mathbf{L}_i$ and $\mathbf{L}_j$ that have the same child sets, denoted as $Ch = \{\mathbf{X}_{\mathbf{t_1}}, \mathbf{X}_{\mathbf{t_2}}, \cdots, \mathbf{X}_{\mathbf{t_m}}, \mathbf{Y}_{t'_1}, \cdots, \mathbf{Y}_{t'_q}\}$, where $\{\mathbf{Y}_{t'_1}, \cdots, \mathbf{Y}_{t'_q}\} \subseteq \{\mathbf{Y}_{\mathbf{t_1}}, \mathbf{Y}_{\mathbf{t_2}}, \cdots, \mathbf{Y}_{\mathbf{t_m}}\}$. Let $\bar{Ch} = \{\mathbf{X}_{\mathbf{t_1}}, \mathbf{Y}_{\mathbf{t_1}}, \cdots, \mathbf{X}_{\mathbf{t_m}}, \mathbf{Y}_{\mathbf{t_m}}\} - Ch = \{\mathbf{Y}_{t'_{q+1}}, \mathbf{Y}_{t'_{q+2}}, \cdots, \mathbf{Y}_{t'_m}\}$.

Notice that the original set $\mathcal{F} = \{\emptyset, Pa(\mathbf{X}_{\mathbf{t_1}}), Pa(\mathbf{Y}_{\mathbf{t_1}}), \cdots, Pa(\mathbf{X}_{\mathbf{t_m}}), Pa(\mathbf{Y}_{\mathbf{t_m}})\}$ can be divided into two distinct partition based on the sets $Ch$ and $\bar{Ch}$. The sets in one partition, $\{\emptyset, Pa(\mathbf{Y}_{t'_{q+1}}), \cdots, Pa(\mathbf{Y}_{t'_m})\} \subset \mathcal{F}$ do not includes either $\mathbf{L}_i$ or $\mathbf{L}_j$, while the sets in the other partition, $\{Pa(\mathbf{X}_{\mathbf{t_1}}), Pa(\mathbf{X}_{\mathbf{t_2}}), \cdots, Pa(\mathbf{X}_{\mathbf{t_m}}), Pa(\mathbf{Y}_{t'_1}), \cdots, Pa(\mathbf{Y}_{t'_q})\} \subset \mathcal{F}$, contains both $\mathbf{L}_i$ and $\mathbf{L}_j$. Therefore, when performing the set subtraction, the result set can either contains both $\mathbf{L}_i$ and $\mathbf{L}_j$, or it can contains neither $\mathbf{L}_i$ nor $\mathbf{L}_j$, both of which still belong to one of the partitions. Hence, it is impossible to generate the singleton $\{\mathbf{L}_i\}$ and $\{\mathbf{L}_j\}$, thus contradicting the assumption "the set of sets $\mathcal{F}$ includes all singleton sets". Consequently, we conclude that the direction $\Rightarrow$ holds.

Next, we will prove the direction "$\Leftarrow$" (i.e., "if"). To begin, let us introduce the property of set subtraction. Consider two sets, denoted as $\mathcal{A}$ and $\mathcal{B}$. Performing set subtraction on these two sets yields three distinct sets: $\mathcal{A} - \mathcal{B}$, $\mathcal{B} - \mathcal{A}$, and $\mathcal{A} - (\mathcal{A} - \mathcal{B})$. Notably, $\mathcal{B} - (\mathcal{B} - \mathcal{A})$ is equal to $\mathcal{A} - (\mathcal{A} - \mathcal{B})$, thus obviating the need to introduce this particular set. Furthermore, it is obvious that $(\mathcal{A} - \mathcal{B}) \cup (\mathcal{B} - \mathcal{A}) \cup (\mathcal{A} - (\mathcal{A} - \mathcal{B})) = \mathcal{A} \cup \mathcal{B}$. And the cardinality of three generated new sets are constrained by: $\min(|\mathcal{A} - \mathcal{B}|, |\mathcal{B} - \mathcal{A}|, |\mathcal{A} - (\mathcal{A} - \mathcal{B})|) \leq \frac{1}{2} \max(|\mathcal{A}|, |\mathcal{B}|)$.

Let us now consider the original set $\mathcal{F} = \{\emptyset, Pa(\mathbf{X}_{\mathbf{t_1}}), Pa(\mathbf{Y}_{\mathbf{t_1}}), \cdots, Pa(\mathbf{X}_{\mathbf{t_m}}), Pa(\mathbf{Y}_{\mathbf{t_m}})\}$. Note that the parent sets of every $\mathbf{X}_{\mathbf{t}}$ contain all the latent factors $\mathbf{L}$, which can be represented as the universal set $\mathcal{U}$. Therefore, we can select one of the $\mathbf{X}_{\mathbf{t}}$, denoted as $\mathbf{X}$, as it encompasses the entire set of latent factors. Next, we consider a total of $m + 1$ observed variables, where $m$ of them are denoted as $\mathbf{Y}_{\mathbf{t}}$, and one of them is $\mathbf{X}$. According to Lemma 3, the cardinality of their parent sets is no more than $2^m$. Here we present a set generating process: Firstly, we have a set $Pa(\mathbf{X})$. Next, by introducing the set $Pa(\mathbf{Y}_{\mathbf{t_1}})$ and performing set subtraction, we obtain three new sets $Pa(\mathbf{X}) - Pa(\mathbf{Y}_{\mathbf{t_1}})$, $Pa(\mathbf{Y}_{\mathbf{t_1}}) - Pa(\mathbf{X})$ and $Pa(\mathbf{X}) - (Pa(\mathbf{X}) - Pa(\mathbf{Y}_{\mathbf{t_1}}))$. Subsequently, we introduce the set $Pa(\mathbf{Y}_{\mathbf{t_2}})$ and perform set subtraction on each of these three sets, resulting in nine new sets. We repeat this process by introducing $Pa(\mathbf{Y}_{\mathbf{t_2}}), \cdots, Pa(\mathbf{Y}_{\mathbf{t_m}})$ and performing set subtraction. Finally, we obtain $3^m$ generated sets denoted as $\mathcal{S}$. As mentioned earlier, the union set of these $3^m$ generated sets is the universal set $\mathcal{U}$. Moreover, the cardinality of these sets is constrained

by the following condition:

$$|\mathcal{S}| \le \frac{1}{2} \cdot \frac{1}{2} \cdots \frac{1}{2} \cdot |Pa(X)| \le (\frac{1}{2})^m \cdot 2^m = 1. \tag{47}$$

Equation 47 indicates that the cardinality of each generated set is no more than 1, implying that they are either empty sets or singletons. Combining this with the fact that the union set is the universal set, we can conclude that $\{\mathbf{L}_1\}, \{\mathbf{L}_2\}, \cdots, \{\mathbf{L}_n\} \in \mathcal{F}$. Therefore, the direction $\Leftarrow$ holds.

□

### A.4 PROOF OF THEOREM 2

**Theorem 2.** *Considering the data generating process described in Section 3.1, we employ a binary adjacency matrix denoted as $A$ to represent the topology relations between $\mathbf{L}_i$ and $\mathbf{Y_t}$. The matrix $A$ comprises $m$ rows and $n$ columns, where $m$ is the number of $\mathbf{Y_t}$, and $n$ is the number of latent factors $\mathbf{L}_i$. Specifically, a value of 1 at position $(i, j)$ indicates that $\mathbf{L}_j$ has a direct effect on $\mathbf{Y}_i$, while a value of 0 indicates no direct effect. Latent factors in a SCM are identifiable if and only if the following equation holds. We refer to the equation as the **uniform identifiability condition (UIC)**.*

$$\mathbb{1}\left(\frac{1}{m}\left[A^\top A + (1-A)^\top(1-A)\right] - I_{n\times n}\right) = 0_{n\times n}. \tag{5}$$

*In Equation 5, $\mathbb{1}(\cdot)$ is an indicator function, which defined as $[\mathbb{1}(A)]_{ij} = \begin{cases} 0, & 0 \le a_{ij} < 1 \\ 1, & a_{ij} = 1 \end{cases}$*

*Proof.* The proof consists of three steps, and an overview of the proof is presented in Figure 5.

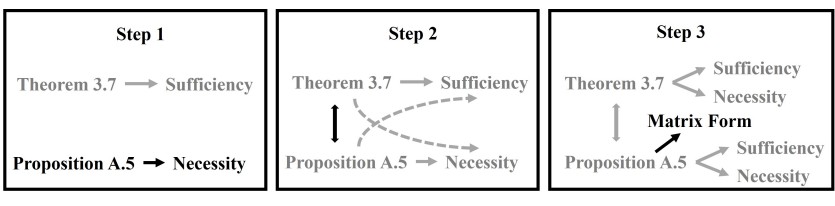

Figure 5: Overview of the proof. Each step focuses on the element marked in black. In Step 1, we demonstrate that the condition stated in Proposition 1 is a necessary condition for determining SCM identifiability. In Step 2, we establish the equivalence between the conditions in Proposition 1 and Theorem 1, thereby showing that both conditions are necessary and sufficient. Finally, in Step 3, we present the matrix form representation of the condition in Proposition 1.

**Step 1. Proving Necessity** We introduce a criterion to determine the identifiability of a given SCM. The criterion is that: For any two distinct latent factors $\mathbf{L}_i$ and $\mathbf{L}_j$ in the SCM, their child sets (i.e., sets containing $\mathbf{X}_t$ and $\mathbf{Y}_t$ pointed by $\mathbf{L}$) are not identical. Then, according to Lemma 1, a negative answer to this criterion implies the non-identifiability of the SCM. Thus it can be seen as the contrapositive form of the necessary condition for identifiability. We can express the equivalent necessary condition in the form of a proposition:

**Proposition 1.** *If the SCM is identifiable, then for any two distinct latent factors $\mathbf{L}_i$ and $\mathbf{L}_j$ in the SCM, their child sets are not identical.*

**Step 2. Combining Necessity and Sufficiency** Notice that Theorem 1 provides a sufficient condition for the identifiability of SCM, while Proposition 1 presents a necessary condition for the identifiability of SCM. According to Lemma 4, these two conditions are exactly equivalent. Consequently, we conclude that both conditions are both necessary and sufficient for the identifiability of SCM. Based on the conclusion, we can strengthen Proposition 1 by incorporating the sufficiency aspect, as presented in Proposition 2.

**Proposition 2.** *A SCM is identifiable, **if and only if** for any two distinct latent factors $\mathbf{L}_i$ and $\mathbf{L}_j$ in the SCM, their child sets are not identical.*

**Step 3. Matrix Representation**    In this step, we will represent the conditions using matrix notation. Notice that the condition described in Theorem 1 involves a generative process, which poses challenges when attempting to express it in matrix form. Therefore, we choose to employ the condition introduced in Proposition 2, i.e., for any two distinct latent factors $\mathbf{L}_i$ and $\mathbf{L}_j$ in the SCM, their child sets are not identical. This condition can be naturally expressed using a binary adjacency matrix denoted as $A$. The matrix $A$ comprises $m$ rows and $n$ columns, where $m$ is the number of $\mathbf{Y_t}$, and $n$ is the number of latent factors $\mathbf{L}_i$. Specifically, a value of 1 at position $(i, j)$ indicates that $\mathbf{L}_j$ has a direct effect on $\mathbf{Y}_i$, while a value of 0 indicates no direct effect. The condition that the child sets are not identical is equivalent to stating that any two distinct columns in matrix $A$ are not the same. Hence, we can express Proposition 2 in matrix form as Proposition 3.

**Proposition 3.** *Considering the binary adjacency matrix A described in Step 3, a SCM is identifiable, **if and only if** any two distinct columns in matrix A are not the same.*

Notice the following Equation 48 holds:

$$x_1 = \{0, 1\}, \quad x_2 = \{0, 1\}, \quad x_1 x_2 + (1 - x_1)(1 - x_2) = \begin{cases} 1 & x_1 = x_2 \\ 0 & x_1 \neq x_2 \end{cases} \tag{48}$$

The formula $x_1 x_2 + (1 - x_1)(1 - x_2)$ can be regarded as a correlation function for $x_1$ and $x_2$, and this correlation function can be straightforward generalized to a vector form:

$$C_{ij} \triangleq Corr(\mathbf{v}_i, \mathbf{v}_j) = \frac{1}{\dim(\mathbf{v})} \left[ (\mathbf{v}_i^\top \mathbf{v}_j) + (1 - \mathbf{v}_i)^\top (1 - \mathbf{v}_j) \right], \tag{49}$$

where the term $\frac{1}{\dim(\mathbf{v})}$ serves as a normalization factor. $C_{ij} = 0$ if all of the elements in the same position of $\mathbf{v}_i$ and $\mathbf{v}_j$ are different. $0 < C_{ij} < 1$ if some of the elements in the same position of $\mathbf{v}_i$ and $\mathbf{v}_j$ are the same. $C_{ij} = 1$ if $\mathbf{v}_i$ and $\mathbf{v}_j$ are exactly the same.

Based on that, we can express the condition that "any two distinct columns in matrix $A$ are not the same" using an equivalent matrix formula, as shown in Equation 5:

$$\mathbb{1} \left( \frac{1}{m} \left[ A^\top A + (1 - A)^\top (1 - A) \right] - I_{n \times n} \right) = 0_{n \times n}.$$

Here the indicator function $\mathbb{1}(\cdot)$ acts as a selector to identify which two columns are identical.    $\square$

## B    PROMPT ENGINEERING

Table 5: Design of discrete prompt described in natural language. For classification tasks, we provide category options as part of prompt.

| Task | Discrete Prompt |
|------|-----------------|
| SUM | Summarize the document: |
| RC | Answer the question based on its following passage: |
| TC | Distinguish which topic the text is (options are [option]): |
| PD | Distinguish whether the two sentences have the same meaning (options are [option]): |
| SA | Distinguish which sentiment the review is (options are [option]): |
| LA | Distinguish whether the sentence is linguistically acceptable (options are [option]): |
| NLI | Distinguish whether the first sentence can infer its following sentence (options are [option]): |

For both Vanilla-IT and SIT, we apply the same setting of prompt engineering as follow.

We adopt hybrid prompts $p = \{p_d, p_c\}$ as instructions following (Xu et al., 2022; Chen et al., 2023), where discrete prompts $p_d$ are natural words, while continuous prompts $p_c$ are continuous embeddings. For the discrete prompts $p_d$, we manually design them as shown in Table 5. For the continuous prompts $p_c$, we utilize an individual prompt encoder to encode a sequence of trainable dense vectors. The prompt encoder is composed of two-layer bidirectional long-short term memory network (BiLSTM) (Graves & Graves, 2012) followed by a multilayer perceptron (MLP), i.e.,

$p_c = \text{MLP}(\text{BiLSTM}([p_1], [p_2], ..., [p_{|p_c|}]))$, where $[p_j]_{j=1}^{|p_c|}$ represents placeholders to be replaced by trainable dense vectors, of length $|p_c| = 6$ for each input sequence. Note that multiple source sequences are concatenated into one as input. In this work, there are at most two source sequences, and the prompted input is $< p, x_1, x_2 > = \{[s], p_d, p_c, x_1, [e], p_c, x_2, [e]\}$ for such tasks.

For the prompt encoder, the mid-hidden size and output size of the LSTM is 512 and 1024, respectively. Dropout with probability 0.1 is applied for LSTM. MLP is composed of two linear layers with a ReLU activation function in between. The hidden size and output size of the two-layer MLP is 1024.

## C  DETAILS OF CAUSAL FACTOR SELECTION

In this section, we introduce the implementation details of the task representation $\mathbf{h}_t$ and the latent mask vector $\mathbf{m}_t$.

**Task Representation**. We obtain task representation $\mathbf{h}_t$ by encoding hybrid prompts $p = \{p_d, p_c\}$ introduced in Appendix B. Specifically, for discrete prompts with variable length, we derive a single embedding $\mathbf{e}_{p_d} \in \mathbb{R}^{d_h}$ through the utilization of average pooling, applied to the output embedding sequence generated from a word embedding layer. Also, for continuous prompts with the maximum length of 12 (the length twice as long as 6 for two source sequences), we linearly transform the output embedding sequence from the prompt encoder into another embedding $\mathbf{e}_{p_c} \in \mathbb{R}^{d_h}$. Then, we linearly combine them to achieve the task representation $\mathbf{h}_t \in \mathbb{R}^n$, i.e., $\mathbf{h}_t = \mathbf{W}_4 \mathbf{e}_{p_d} + \mathbf{W}_5 \mathbf{e}_{p_c} + \mathbf{b}_4$.

**Latent Mask Vector**. We obtain the latent mask vector $\mathbf{m}_t$ based on the task representation $\mathbf{h}_t$. Firstly, $\mathbf{h}_t$ is normalized by a sigmoid activation function into $\hat{\mathbf{h}}_t$, a soft version of latent mask vector, i.e.,

$$\hat{\mathbf{h}}_t = \text{Sigmoid}(\mathbf{h}_t), \tag{50}$$

whose continuous value $\hat{h}_{ti} \in (0, 1)$ in each dimension represents the selected probability of each latent factor. Then, we utilize bernoulli sampling to obtain the hard latent mask vector $\mathbf{m}_t$ according to $\hat{\mathbf{h}}_t$, where the discrete value $m_{ti} \in \{0, 1\}$ in each dimension is sampled from $\{0, 1\}$ and only 1 represents "selected". To increase the stability of sampling results, we additionally multiply a scaling coefficient selected from $(50, 200)$ for $\mathbf{h}_t$ before the sigmoid activation.

## D  DETAILS OF CAUSAL FACTOR CONSTRAINT

**UIC Loss Function**. Note that Equation 5 provides a necessary and sufficient condition for identifying latent factors. Using this equation, we can design a loss function to ensure identifiability in our model. However, a challenge arises with the indicator function in Equation 5, which is non-differentiable at $a_{ij} = 1$. This prevents direct application of gradient-based optimization methods. One solution is to replace the indicator function with an approximate, but differentiable function. In this work, we choose the power function $x^\alpha$, which becomes increasingly similar to the indicator function as $\alpha$ approaches infinity (Practically, $\alpha = 50$ is quite enough) . Therefore, our loss function can be expressed as:

$$\mathcal{L}_{uic} = \frac{1}{m^\alpha} \sum_{i,j=1}^{n} \left[ \sum_{k=1}^{m} a_{ki} a_{kj} + (1 - a_{ki})(1 - a_{kj}) - m \delta_{ij} \right]^\alpha, \tag{6}$$

In Equation 6, $\delta$ is the Kronecker delta function, which defined as $\delta_{ij} = \begin{cases} 0, & i \neq j \\ 1, & i = j \end{cases}$

**Implementation of Matrix** $A$. To apply the UIC loss and task distinction loss, we construct a discrete task-latent matrix $\mathbf{M}_{tl}$ to implement the binary adjacency matrix $A$ described in Theorem 2, whose elements $a$ are utilized in $\mathcal{L}_{uic}$ (Equation 6) and $\mathcal{L}_{dis}$ (Equation 7).

First, we construct a continuous version of this matrix. Specifically, we collect the soft latent mask vectors $\hat{\mathbf{h}}_t \in \mathbb{R}^n$ (introduced in Appendix C) for $m$ training mixture tasks, and stack $m$ vectors into a continuous matrix $\mathbf{M} \in \mathbb{R}^{m*n}$. Collected from training batches, this matrix changes dynamically.

Then, we discretize this continuous matrix. Since the bernoulli sampling does not meet the requirement of derivability, we apply gumbel-softmax rather than bernoulli sampling to realize discretization of $\mathbf{M}_{tl}$ for parameter optimization during training.

# E    DETAILS OF TASKS AND DATASETS

In this section, we present the task selection as well as our sampling strategy.

**Task Selection**. We selected tasks based on established prior works like FLAN and T0, choosing widely-adopted datasets to validate our approach. This selection covers a diverse range of classical NLP tasks, including the General Language Understanding Evaluation (GLUE) benchmark, which is one of the most popular evaluation benchmarks used for multitask learning (Worsham & Kalita, 2020). Besides Natural Language Understanding (NLU) tasks, we also consider Natural Language Generation (NLG) tasks, e.g., Summarization and Reading Comprehension.

The training datasets consist of XSUM (Narayan et al., 2018), CNNDM (See et al., 2017), $\text{Duorc}_{self}$ (Saha et al., 2018), $\text{Duorc}_{para}$ (Saha et al., 2018), AG (Zhang et al., 2015), Trec (Li & Roth, 2002), PAWS (Zhang et al., 2019), IMDB (Maas et al., 2011) and Yelp (Zhang et al., 2015). The held-out datasets used are Gigaword (Napoles et al., 2012), Squad (Rajpurkar et al., 2016), DBPedia (Lehmann et al., 2015), MRPC (Dolan & Brockett, 2005), QQP (Wang et al., 2018), SST-2 (Socher et al., 2013), CoLA (Warstadt et al., 2019), $\text{MNLI}_m$ (Williams et al., 2018), $\text{MNLI}_{mm}$ (Williams et al., 2018), QNLI (Rajpurkar et al., 2018), RTE (Dagan et al., 2006), WNLI (Levesque et al., 2012). Details of all datasets are provided in Table 6a.

**Sampling Strategy**. To construct the training mixture dataset, we randomly sample and mix data from each dataset listed in Table 6a. Following the approach described in (Wei et al., 2021a; Raffel et al., 2020), we adopt the examples-proportional mixing scheme and limit the number of training examples per dataset to 15k. In order to increase the coverage of the sampled dataset with respect to the original dataset, we prioritize sampling data that has not been sampled before. Consequently, the sample size of the training mixture datasets in our work can be expressed as:

$$\text{size} = \min\left(\text{num(epochs)} \times 15\text{k}, \text{size(original dataset)}\right), \qquad (51)$$

where the number of training epochs is 10 in our works. The statistics of the final training mixture datasets and the held-out datasets are shown in Table 6.

Table 6: Data statistics.

(a) The training mixture datasets.

| Task | Dataset | Train (sampled) | Test |
|------|---------|-----------------|------|
| **SUM** | XSUM | 150,000 | 11,334 |
| | CNNDM | 150,000 | 11,490 |
| **RC** | $\text{Duorc}_{self}$ | 60,721 | 12,559 |
| | $\text{Duorc}_{para}$ | 69,524 | 15,857 |
| **TC** | AG | 120,000 | 7,600 |
| | Trec | 5,452 | 500 |
| **PD** | PAWS | 49,401 | 8,000 |
| **SA** | IMDB | 25,000 | 25,000 |
| | Yelp | 150,000 | 50,000 |

(b) The held-out datasets.

| Task | Dataset | Split | Size |
|------|---------|-------|------|
| **SUM** | Gigaword | test | 1,951 |
| **RC** | Squad | dev | 10,570 |
| **TC** | DBPedia | test | 70,000 |
| **PD** | MRPC | dev | 408 |
| | QQP | dev | 40,430 |
| **SA** | SST-2 | dev | 872 |
| **LA** | CoLA | dev | 1,043 |
| **NLI** | $\text{MNLI}_m$ | dev | 9,815 |
| | $\text{MNLI}_{mm}$ | dev | 9,815 |
| | QNLI | dev | 5,463 |
| | RTE | dev | 277 |
| | WNLI | dev | 71 |

## F  DETAILS OF TRAINING AND INFERENCE

In this section, we supplement more details about the training and inference process. For the tasks with one source sequence, we set the max length as 550, while for those with two source sequences, we set the max length as 350 for the first sentence, and 200 for the second sentence. For other hyper-parameters, we manually tune them based on the validation set or a subset of training set. Specifically, the batch size is selected from $\{256, 512\}$, the learning rate is selected from $\{1e^{-5}, 3e^{-5}, 5e^{-5}\}$. The total training steps contain 10 epochs, and we conduct evaluation for early stopping every epoch and every 500 steps. During inference, we apply beam search for text generation and set beam size as 6. Specifically, we use Huggingface Transformers library [4] for implementations [5]. All the reported results come from evaluation on models trained in the mixture datasets, which are subsets sampled from the full datasets.

## G  FEW-SHOT LEARNING

In this section, we show all the experimental results under the few-shot setting in Table 7. The hyper-parameter setup is the same as the setup during training stage, except for the warm-up strategy absent in few-shot training. The last checkpoint are picked for prediction.

As shown in Table 7, SIT outperforms Vanilla-IT on 9 out of 12 datasets, demonstrating the better learning capability and generalization ability of SIT, which benefits from SCM capturing the underlying causal relationships. On the whole, the model performance improves more on the difficult tasks after few-shot learning, e.g. SUM task, while the performance on the simple tasks maybe decrease, e.g., RC task. We analyze the two cases in detail as follows. (i) On the datasets that have poor zero-shot performance, e.g., DBPedia and CoLA, both Vanilla-IT and SIT gain significantly under the few-shot setting as shown in Figure 2. The larger gain of SIT than Vanilla-IT indicates that structural instructions can adapt faster and better to a new target space with SCM as bridge between the task and target. (ii) On the datasets that have good zero-shot performance, e.g., SST-2, Vanilla-IT can only improve 0.87% in terms of accuracy by learning few samples, while SIT leads to a decrease in model performance. The possible reason is that with $3e^{-5}$ as learning rate the same as training stage, the update rate of the model parameters is too fast, so that the prediction behavior is unstable or even worse for the tasks previously performed well. More suitable hyper-parameter setup needs to be determined by grid search.

Table 7: Few shot performance of all the held-out datasets, including OOD and cross-task situations.

| Method | OOD Performance | | | | | Cross-task Performance | | | | | | |
| | SUM | RC | TC | PD | | SA | LA | NLI | | | | |
| | Gigaword | Squad | DBPedia | MRPC | QQP | SST-2 | CoLA | MNLI$_m$ | MNLI$_{mm}$ | QNLI | RTE | WNLI |
| --- | --- | --- | --- | --- | --- | --- | --- | --- | --- | --- | --- | --- |
| Vanilla-IT | 29.82 | 54.02 | 76.33 | 68.38 | 36.82 | **93.23** | 38.16 | 32.65 | 32.94 | 50.52 | 16.97 | 43.66 |
| SIT | **30.05** | **75.99** | **93.16** | 68.38 | **74.52** | 87.96 | **69.03** | **35.39** | **35.21** | **50.54** | **47.29** | 43.66 |

---

[4]https://github.com/huggingface/transformers
[5]The code is available at https://anonymous.4open.science/r/SIT-34DB/

