# OpenReview forum: "A Unified Causal View of Instruction Tuning"
_ICLR.cc/2024/Conference — Submitted to ICLR 2024_

### Official Review · Reviewer_GNKT · 2023-10-30

**Soundness:** 3 good
**Presentation:** 3 good
**Contribution:** 3 good
**Rating:** 6
**Confidence:** 3

**Summary:**

This paper presents a unified causal view of instruction tuning in natural language processing (NLP) tasks. The authors propose a meta Structural Causal Model (meta-SCM) that integrates different NLP tasks under a single causal structure. They introduce task-required causal factors and develop a Structural Instruction Tuning (SIT) method to learn the task-required causal representations. The effectiveness of their approach is demonstrated through improvements in zero-shot learning on unseen datasets and tasks.

**Strengths:**

1. The paper introduces a novel instruction tuning method based on a meta-SCM, which captures underlying causal relationships and enhances adaptability to domain transfer and new tasks.
2. This paper proposes a novel Uniform Identifiability Condition (UIC) based on the topology of Structural Causal Model and the rationality of UIC is verified with a detailed mathematical proof.
3. The paper provides detailed experimental results and comparisons with baselines, demonstrating the superior performance of the proposed method in terms of in-domain, out-of-domain, and cross-task datasets.

**Weaknesses:**

1. This paper should compare the performance of their methods with different baselines and backbones to comprehensively validate the effectiveness of proposed method.
2. In the experiment, this paper should give some evidence that the model learn causal representations and task-oriented causal generative mechanisms.

**Questions:**

No questions.

---

> ### Author Response · Authors · 2023-11-21
> **Response to Reviewer GNKT**
>
> We thank the reviewer for the constructive feedback. As suggested, we will compare our method to additional baselines using different backbones to further validate its effectiveness. We will also analyze the representations learned by our model and provide supporting evidence that causal mechanisms are captured.
>
> We also sincerely appreciate the reviewer highlighting our key contributions - the novel meta-SCM, Uniform Identifiability Condition proof, and detailed experimental comparisons. Your feedback has been invaluable for improving our work. Please let us know if you have any other suggestions for strengthening the paper. Thank you again for your time in thoroughly reviewing our submission and providing these thoughtful suggestions.

---

### Official Review · Reviewer_nrtp · 2023-10-31

**Soundness:** 1 poor
**Presentation:** 2 fair
**Contribution:** 2 fair
**Rating:** 5
**Confidence:** 2

**Summary:**

This paper tries to learn task-required causal factors and only use those to make predictions for a given task. The motivation is claimed to be that the causal factors of different tasks are spuriously correlated through the task variable $T$. The authors theoretically prove the causal factor can be identified without mixing information from others, then they propose a Structural Instruction Tuning (SIT) method to learn the task-required causal representations that can mimic the causal factors for each task.

**Strengths:**

1. Exploring causal LLMs is very important for both research and applications.

2. This is a very novel work in this direction.

**Weaknesses:**

1. The organization of this paper is very hard to follow.

2. The technical quality is poor. There are serious technical flaws in this paper. See the questions below.

**Questions:**

Q1. SCMs should be constructed from structual functions. Intuitive causal graphs are very informal and not relible. Equation 1 is not enough, because the structual functions should appear before the graphs. I think it would be better to illustrate Fig 1 with a specific example (i.e., what $L_i, X, Y$ and $T$ stands for in a specific task, and how the non-causal factors correlate with $Y_t$).

Q2. The authors use different $L_i$ for different causal factors, and this is good. Nevertheless, I think $Y_t$ in Fig.1 should also be notated as different nodes for different $t$.

Q3. The authors claim that "Causally speaking, there
exist backdoor paths between the target labels and non-causal latent factors through the task
variable T.". But when we are doing a task, the task variable is given. And in Fig.1, we have $L_i \perp L_j|T$. So, there must be a problem arising from either the claim itself or the causal graph. Considering question Q1, I think it is more likely the latter. I would appreciate it if the authors could provide a specific example to illustrate how the non-causal factors can correlate with $Y_t$ (e.g., the simplest text classification). This is one of the most important questions for me, and I'm looking forward to the authors' response.

Q4. Following the above question, even if $T$ can act as a confounder and spurious correlations exist in the data, how does the traditional training process make the model learn the spurious correlations? This problem is also untouched, which makes the motivation informal and weak.

Q5. It seems that the only confounder in this paper is the task $T$, I can hardly come up with any applications where different features are caused by the task but don't cause the target label.

Q6. Section 3.2 is really hard to follow. How to alleviate the influence of the spurious correlations? By backdoor adjustment or introducing intervened data?

Q7. The definition of identifiability is confusing. What's the identifiability of a SCM? When we say the identifiability, it usually refers to the causal quantity on two specific variables rather than a SCM. Also, the target of the UIC loss in Eq.6 is vague. What causal quantity do you want to measure?

Q8. I guess the authors want to say that, if $L_i$ is identifiable, then it is the causal factor of $Y_t$. So, the UIC loss is used to promise the identifiability. There are two issues need to be addressed. One is why identifiable factors are causal ones, and another is why UIC can select identifiable factors. I'm not clear whether the second problem has been addressed by Section 3.2, since I think it's not well orangized. Nevertheless, I think the first problem is untouched.

Q9. I'm not sure whether the baselines are advanced enough. The paper only compares with two simple baselines.

---

> ### Author Response · Authors · 2023-11-21
> **Response to Reviewer  nrtp (1/2)**
>
> We deeply appreciate your careful, detailed review. Your thorough feedback is valuable for revising our paper - we have re-uploaded the modified version highlighting major changes in blue. The questions you ask are critical, and they all involve the core of our research. We are honored to discuss them with you.
>
> Before answering your questions, we first provide a driven example, which we hope will intuitively show the motivation for our modeling, the potential cause of spurious correlation, and our solution:
>
> **Driven example:**
>
> Let's consider the sentence "Pizza is delicious." This is an instance of the random variable $X_t$ in our paper. This sentence contains information about food, "pizza", and sentiment, "delicious." Accordingly, we introduce latent factors to represent different types of information. Specifically, $ L_1$ represents food information and $L_2$ represents sentiment. Thus, we have $L_1, L_2 \rightarrow X_t$ . We use $Y_t$ to represent the target labels of different tasks, which require different latent factors, such as $L_1$ for topic classification and $L_2$ for sentiment classification.
> Due to inherent dataset properties (probably from sampling bias), spurious correlation may arise. For example, in a dataset sampled from pizza enthusiasts for sentiment classification, pizza, as a food concept, will frequently co-occur with positive emotion, causing spurious correlation between food and sentiment labels.
> In our paper, we intended $D$ to represent inherent dataset properties and thus $D \rightarrow L_1, L_2$ . Since $D$ denotes internal properties, it is unobservable. For $Y_t$ of sentiment classification, there is a backdoor path between $Y_t$ and $L_1$, i.e., $L_1 \leftarrow D \rightarrow L_2 \rightarrow Y_t$ , producing spurious correlation between food information $L_1$ and positive sentiment labels $Y_t$ .
>
> **Unclear statement of $T$ and our revision:**
>
> In our original paper, for the sake of narrative convenience, we imprecisely used $T$ to denote both inherent dataset properties and tasks. We apologize for any confusion caused by this unclear statement. We sincerely appreciate you raising this critical issue in Q3, alerting us to this imprecision in our writing. We have already revised the paper where we use $D$ for inherent dataset properties and $T$ for tasks separately.
>
> **Answers for Questions:**
>
> **A1** ：
>
> We agree with you that it is more rigorous to model SCM from the structural equations, and when the structural equations are given, the causal graph will be determined accordingly. The reason we introduced causal graph firstly is that we want to give the reader an intuitive understanding of the data generating process of NLP tasks. This includes introducing the variables and their qualitative causal relationships. An illustrative example corresponding to Fig. 1 has been provided in the above driven example. Thank you very much for your advice.
>
> **A2：**
>
> Thanks for your valuable suggestion. In our modeling, $Y_t$ for different tasks are actually different nodes. For visual simplicity, we depicted only a single $Y_t$ node in the causal graph, and distinguished different tasks by subscript $t$.
>
> **A3：**
>
> This is an excellent question and we appreciate you raising this important issue. $T$ should be modified to $D$ to represent inherent dataset properties, which are unobservable. This would make $L_i$ not independent of $L_j$, leading to a backdoor path and inducing spurious correlation. We will correct the imprecision in our writing, which improperly used $T$ to denote both dataset properties and tasks. Thank you again for catching this critical misunderstanding.

---

> ### Author Response · Authors · 2023-11-21
> **Response to Reviewer  nrtp (2/2)**
>
> **Answers for Questions:**
>
> **A4：**
>
> We appreciate you raising this excellent point. The traditional training process optimizes the likelihood $P(Y|X)$, where $X$ contains a mixture of information - some factors causally influence $Y$ while others are spuriously correlated. Without explicitly disentangling the latent factors within $X$, the shortcuts from non-causal variables to $Y$ may be exploited, leading the model to learn these unintended spurious correlations along with the true causal relationships.
>
> **A5：**
>
> The $T$ in the original paper should be replaced by $D$, which represents the inherent dataset properties. It can describe the spurious correlation between the target label and the feature (latent factor) by modeling the backdoor between them. We apologize for this misunderstanding because of our unclear writing.
>
> **A6：**
>
> Since both the inherent dataset properties $D$ and latent factors are unobservable, we cannot directly perform backdoor adjustment or intervention. Therefore, to mitigate spurious correlation, an alternative approach is to only utilize the parent latent factors of each task's $Y_t$ for prediction.  This branch of research can be categorized as causal representation learning [1]. However, being able to select the parent factors relies on latent factor identifiability. Thus, guaranteeing identifiability of latent factors is a core challenge in causal representation learning.
>
> [1] Towards Causal Representation Learning. Proceedings of the IEEE, 2021.
>
> **A7：**
>
> That is an excellent question. In causal inference, identifiability refers to " A causal quantity can be computed from a purely statistical quantity." [2]. However, in causal representation learning, identifiability has a different meaning - it indicates that "Representations for each latent factor can be learned without mixing information with others, while ensuring that the difference between the learned representations and the true representations remains within acceptable bounds of uncertainty" [3, 4, 5]. The goal of the UIC is to guarantee latent variable identifiability under this definition from causal representation learning.
>
> [2] Introduction to causal inference. 2020.
>
> [3] Self-supervised learning with data augmentations provably isolates content from style. NeurIPS 2021.
>
> [4] Partial Identifiability for Domain Adaptation. ICML 2022.
>
> [5] Identifiability results for multimodal contrastive learning. ICLR 2023.
>
> **A8：**
>
> Thank you for your insightful question. We apologize that Section 3.2 did not effectively communicate our approach regarding identifiability and causal factor selection.
>
> To clarify, we do not posit that "if $L_i$ is identifiable, then it is a causal factor of $Y_t$." For each task, whether $L_i$ is a causal parent of $Y_t$ is an intrinsic property. In our work, causal factors for different tasks are selected by a binary matrix with 0,1 values. This matrix is learned by maximizing likelihood, similar to score-based causal discovery [6]. Notably, selecting causal factors relies on the premise that the factors $L_i$ are identifiable. Thus, we introduce a UIC Loss as a regularization term to ensure $L_i$ identifiability. While UIC constrains the structure between $L_i$ and $Y_t$, this constraint is flexible. Causal factor selection is still primarily driven by the binary selection matrix.
>
> [6] Generalized score functions for causal discovery. SIGKDD 2018.
>
> **A9：**
>
> Thank you for your questions, we will compare our method to additional baselines using different backbones to further validate its effectiveness.
>
> We sincerely appreciate you taking the time to ask such insightful questions. It is clear you want to help improve the quality of our paper, for which we are truly grateful. Please feel free to continue the discussion with any other questions you may have. We welcome the opportunity to further clarify and strengthen our work. Thank you again for your dedication and effort in reviewing our submission so thoroughly.

---

> > ### Comment · Reviewer_nrtp · 2023-11-22
> > **Rating update**
> >
> > The authors have revised the formulation of the questions in their paper, which seems reasonable to a certain degree. However, the correctness of these adjustments still requires further confirmation. There's also a minor issue: the authors mentioned that additional baselines would be introduced, but this appears to have been overlooked in the revised version of the paper. Reflecting on these adjustments, I have decided to increase my rating to 5, while reducing my confidence level to 2.
> >
> > Additionally, I've come across some articles that are notably clear and easy to comprehend. Both [1] and [2] demonstrate effective methods for establishing a solid causal relationship between variables. The authors may find it beneficial to cite and refer to these works in their final, camera-ready revision to enhance the question's framing and the paper's overall readability.
> >
> > References:
> > [1] 'Desiderata for Representation Learning: A Causal Perspective,' arXiv:2109.03795.
> > [2] 'D-Separation for Causal Self-Explanation,' NeurIPS 2023.

---

> > > ### Author Response · Authors · 2023-11-23
> > > **Thank you for engaging with our work and increasing the rating.**
> > >
> > > Dear Reviewer nrtp,
> > >
> > > We sincerely appreciate you taking the time to thoroughly review our paper and offer such valuable feedback. Incorporating your insights certainly helped refine and improve the quality of our manuscript.
> > >
> > > We also sincerely appreciate you increasing the rating. We look forward to the formal rating modification being applied as well.
> > >
> > > With gratitude and warmest regards,
> > >
> > > Authors

---

> ### Author Response · Authors · 2023-11-22
> **Thanks for your valuable feedback**
>
> We deeply appreciate the reviewer kindly increasing the rating and providing constructive feedback to improve our work.
>
> Due to computational constraints, the scale of our current experiments is limited. We will compare to additional baselines using different backbones to further validate effectiveness of our method.
>
> Thank you for recommending articles [1-2] – drawing insights from both, we have already revised our paper to cite them. We will continue enhancing the question framing and improving overall readability referring to these works.
>
> We are truly grateful for your continual thoughtful suggestions, which significantly strengthen our paper’s quality. Thank you again sincerely for the invaluable advice throughout the review process!
>
> [1] 'Desiderata for Representation Learning: A Causal Perspective,' arXiv:2109.03795.
>
> [2] 'D-Separation for Causal Self-Explanation,' NeurIPS 2023.

---

### Official Review · Reviewer_6m14 · 2023-11-01

**Soundness:** 3 good
**Presentation:** 3 good
**Contribution:** 3 good
**Rating:** 6
**Confidence:** 1

**Summary:**

This paper proposes a causal framework to identify latent factors on the properties of a task, and only use these task-related factors to make predictions. It proposes a structural instruction tuning method to learn the representations for each task and demonstrate its effectiveness.

**Strengths:**

1. It is intuitively correct to consider the task-related factors and discard spurious correlations that might lead to vulnerable predictions

**Weaknesses:**

1. I don't fully understand the paper and cannot judge the weakness correctly. But, it looks like from the experimental setup, the scale of experiments being conducted is much smaller than the current NLP benchmarks on instruction tuning. The tasks are carefully chosen and the model tested is not a SOTA seq2seq model that widely adopted for Instruction Tuning.

**Questions:**

1. I don't have good questions for this paper and will inform the AC to seek information from other reviewers.

---

> ### Author Response · Authors · 2023-11-21
> **Response to Reviewer 6m14**
>
> We sincerely thank the reviewer for recognizing the intuitive merit of considering task-related factors and discarding spurious correlations. Due to computational constraints, the scale of our current experiments is limited. Per the reviewer's suggestions, we will explore scaling up our approach using larger models, such as Llama. We will also expand our method to additional tasks and datasets to further demonstrate its effectiveness. We deeply appreciate the reviewer providing this invaluable feedback to strengthen our work.

---

### Meta-Review · Area_Chair_Wx9F · 2023-12-09

**Metareview:**

This paper presents a causal view of instruction tuning by proposing meta Structural Causal Model (meta-SCM) that integrates different NLP tasks under a single causal structure. It tries to learn task-required causal factors and only use those to make predictions for a given task. The assumption is that the causal factors of different tasks are spuriously correlated through the task variable. The authors theoretically prove the causal factor can be identified without mixing information from others, then they propose a Structural Instruction Tuning (SIT) method to learn the task-required causal representations that can mimic the causal factors for each task. The paper seems not easy to follow, due to the paper organization and a few confusing places/formulations. Some formulations were adjusted after discussion, whose correctness needs further confirmation. More analyses are desired, such as larger-scale experiments (e.g., with Llama), more comparisons with different baselines and LM backbones, more evidence of learned causal representations, etc.

**Justification For Why Not Higher Score:**

- paper not easy to follow due to paper organization and confusing places
- more analyses needed

**Justification For Why Not Lower Score:**

- causal interpretation of instruction tuning
- proposed meta SCM approach, showing effectiveness in certain settings

---

### Decision · Program_Chairs · 2024-01-16

Reject